# Differential roles of lysosomal cholesterol transporters in the development of *C. elegans* NMJs

Amin Guo[1],*, Qi Wu[1],*, Xin Yan[2], Kanghua Chen[1], Yuxiang Liu[1], Dingfa Liang[3], Yuxiao Yang[1], Qunfeng Luo[1], Mingtao Xiong[4], Yong Yu[5], Erkang Fei[4], Fei Chen[1]

**Cholesterol homeostasis in neurons is critical for synapse formation and maintenance. Neurons with impaired cholesterol uptake undergo progressive synapse loss and eventual degeneration. To investigate the molecular mechanisms of neuronal cholesterol homeostasis and its role during synapse development, we studied motor neurons of *Caenorhabditis elegans* because these neurons rely on dietary cholesterol. Combining lipidomic analysis, we discovered that NCR-1, a lysosomal cholesterol transporter, promotes cholesterol absorption and synapse development. Loss of *ncr-1* causes smaller synapses, and low cholesterol exacerbates the deficits. Moreover, NCR-1 deficiency hinders the increase in synapses under high cholesterol. Unexpectedly, NCR-2, the NCR-1 homolog, increases the use of cholesterol and sphingomyelins and impedes synapse formation. NCR-2 deficiency causes an increase in synapses regardless of cholesterol concentration. Inhibiting the degradation or synthesis of sphingomyelins can induce or suppress the synaptic phenotypes in *ncr-2* mutants. Our findings indicate that neuronal cholesterol homeostasis is differentially controlled by two lysosomal cholesterol transporters and highlight the importance of neuronal cholesterol homeostasis in synapse development.**

## Introduction

The establishment of synaptic connections in the nervous system is important for the efficient transmission of information and the proper functioning of neural circuits. At synapses, presynaptic vesicle fusion machinery and postsynaptic receptors are anchored by respective scaffold proteins, which are physically linked to the intracellular cytoskeleton and the trans-synaptic cell-adhesion molecules (1, 2, 3, 4). Synapses can be organized through trans-synaptic signaling pathways that are mediated by cell-adhesion molecules such as neurexins and their ligands (5, 6, 7, 8, 9, 10). Non-neuronal cells that are not part of pre- or postsynaptic specialization can also contribute to synapse formation through direct contact or the secretion of factors (11). One of these secreted factors is cholesterol, which is an essential component of both presynaptic vesicles and postsynaptic lipid rafts and plays crucial roles in neurotransmission and synaptic signaling (12, 13). Synapses primarily develop after astrocyte differentiation (14, 15), and it has been shown that cholesterol derived from the astrocyte significantly promotes the formation and maturation of synapses in vitro (16, 17, 18). Astrocytic cholesterol is also critical for maintaining cholesterol homeostasis in neurons. Astrocytes secrete apolipoprotein E (ApoE) particles containing cholesterol, which are subsequently taken up by neurons via receptors belonging to the low-density lipoprotein receptor family (19, 20, 21). Apart from cholesterol, ApoE particles carry a variety of microRNAs that actively repress cholesterol biosynthesis in neurons (22). Disturbances in the cholesterol synthesis and secretion in astrocytes have been associated with synaptic deficits and neurodegenerative disorders (21, 23, 24, 25, 26, 27, 28, 29). Despite the importance of intercellular crosstalk, our knowledge about the intracellular mechanisms controlling cholesterol homeostasis in neurons and its role in synapse development remains limited.

A few studies have shown that molecules associated with cholesterol synthesis, uptake, and turnover in neurons can influence synapse formation or maintenance (26, 30, 31). Although postnatal cerebellar granule cells deficient in cholesterol biosynthesis develop normal synapses with Purkinje cells, hippocampal neurons deficient in cholesterol biosynthesis form fewer synapses when cultured with a minimal presence of glial cells (30, 32). It is possible that cultured neurons develop non-natural synaptic connections, or their requirements for cholesterol synthesis vary depending on brain regions or the age of the animals from which they are obtained (33). Deletion of low-density lipoprotein receptor–related protein 1 (LRP1) in forebrain neurons of mice results in a decrease in brain cholesterol levels and a progressive, age-dependent decline in synapses (26). The overexpression of

---

[1]School of Basic Medical Sciences, Jiangxi Medical College, Nanchang University, Nanchang, China   [2]School of Life Sciences, Nanchang University, Nanchang, China   [3]Queen Mary School of Nanchang University, Jiangxi Medical College, Nanchang, China   [4]Institute of Biomedical Innovation, Jiangxi Medical College, Nanchang University, Nanchang, China   [5]State Key Laboratory of Cellular Stress Biology, School of Life Sciences, Faculty of Medicine and Life Sciences, Xiamen University, Xiamen, China

Correspondence: feichen@ncu.edu.cn
*Amin Guo and Qi Wu contributed equally to this work

CYP46A1, a 24-hydroxylase that converts excess cholesterol to 24S-hydroxycholesterol, in hippocampal neurons alleviates spine loss in a mouse model with Alzheimer's disease (31). These in vivo studies have implicated the critical role of neuronal cholesterol homeostasis in synapse maintenance. Nevertheless, the causal link between impaired cholesterol homeostasis and synaptic deficits has yet to be established.

Neurons rely on de novo cholesterol biosynthesis for their survival and growth during early development in addition to the glial supply of cholesterol in later stages (30, 32). In rodent brains, inactivation of cholesterol synthesis in postnatal neurons is required for studying the regulation of cholesterol homeostasis during synapse development. However, simply using Cre or Cre-ER to induce gene inactivation does not guarantee complete deletion of cholesterol synthesis genes before astrocyte differentiation, nor does it eliminate the residue cholesterol synthesized in embryonic neurons. Therefore, how exogenous cholesterol achieves homeostasis within neurons to ensure proper synapse development remains an open question. We sought to determine first whether cholesterol homeostasis is important and how it is important in *Caenorhabditis elegans*. Neurons in *C. elegans* cannot de novo synthesize sterols as they lack genes for squalene synthase and cyclase (34), and they require dietary cholesterol for development (35). The use of *C. elegans* would enable us to investigate cholesterol homeostasis regulation in neurons and its role in synapse development.

After endocytosis, cholesteryl esters undergo hydrolysis in lysosomes, yielding unesterified cholesterol that is subsequently exported to various cellular compartments (36). The NPC1 protein, a multiple transmembrane protein located on LE/Ly, is primarily responsible for the egress of cholesterol from lysosomes to other cellular compartments in mammals (37, 38, 39). The *C. elegans* genome encodes two NPC1 homologs, *ncr-1* and *ncr-2*, known to participate in intracellular cholesterol processing, hormone production, and reproductive growth (35, 40, 41, 42). In this study, we first examined cholesterol levels and neuromuscular junctions (NMJs) in *ncr-1* and *ncr-2* null mutants. NCR-1 promotes cholesterol absorption and the development of NMJs. *ncr-1(0)* mutants exhibit smaller NMJs. A low-cholesterol diet exacerbates the deficits of NMJs in *ncr-1(0)* mutants, suggesting that reduced cholesterol absorption causes these deficits. Furthermore, *ncr-1(0)* inhibits the increase in NMJs induced by a high-cholesterol diet, indicating that impaired cholesterol absorption hinders synapse formation. Surprisingly, NCR-2, another homolog of NPC1 in *C. elegans*, facilitates using cholesterol and sphingomyelins and inhibits the formation of NMJs. *ncr-2(0)* mutants exhibit an equivalent increase in NMJs under both low- and high-cholesterol conditions, implying that reduced cholesterol use is sufficient to promote synapse formation. The elevated levels of sphingomyelins contribute to the phenotypes of NMJs in *ncr-2(0)* mutants. Our findings demonstrate the critical role of neuronal cholesterol homeostasis in synapse formation. Cholesterol homeostasis is controlled by two NCR proteins: NCR-1 absorbs cholesterol and promotes synapse development, whereas NCR-2 uses cholesterol and hinders synapse development.

# Results

## NCR-1 promotes cholesterol absorption and synapse development

NCR-1, a cholesterol transporter in *C. elegans*, contains all the structure domains of NPC1 (40), and its protein sequences share 29% identity and 61% similarity to human NPC1 (Figs 1A and S1). To investigate the potential role of NCR-1 in maintaining cholesterol homeostasis and synapse development, we used *ncr-1(nr2022)*, a mutant obtained via an EMS screen (43). The *ncr-1(nr2022)* allele results in the deletion of multiple exons, rendering it null, and is therefore denoted as *ncr-1(0)* (Fig 1A). We first evaluated the impact of *ncr-1(0)* on the lipid composition of *C. elegans* by performing biochemical extraction, followed by mass spectrometry (Fig S3A and B and Supplemental Methods). Although there were no notable changes in the total lipid levels among the 11 subcategories (sphingomyelins (SMs), triacylglycerols (TAGs), diacylglycerols (DAGs), free fatty acids (FFAs), phosphatidylcholines (PCs), phosphatidic acids (PAs), phosphatidylserines (PSs), phosphatidylinositols (PIs), phosphatidylethanolamines (PEs), cardiolipins (CLs), and phosphatidylglycerols (PGs)) between the WT and *ncr-1(0)* mutants (Figs 1B and S3C–L), the *ncr-1(0)* mutants exhibit lower levels of free cholesterol and higher levels of cholesteryl esters compared with the WT (Fig 1C and D). These results suggest that NCR-1 promotes cholesterol mobilization.

We then examined the synapses between GABAergic motor neurons and muscle cells. The vesicle-associated membrane protein SNB-1-GFP/VAMP, driven by the *unc-25* promoter, labels the presynaptic bouton in GABAergic motor neurons (44, 45). Our analysis was focused on the presynaptic boutons present in the dorsal processes of DD motor neurons (Fig 1E). Our observations revealed a significant reduction in the bouton size of *ncr-1(0)* mutants, which was 73% of the size observed in WT animals (Fig 1F and G), whereas the number of GABAergic presynaptic boutons remained constant in *ncr-1(0)* mutants (Fig 1H). In addition, the presynaptic boutons in *ncr-1(0)* mutants displayed a strong co-localization with *UNC-49B-tagRFP* (Fig 1F), which is a subunit of ionotropic GABA receptors (46, 47). These findings indicate that NCR-1, a *C. elegans* homolog of human NPC1, increased cholesterol absorption and facilitated synapse development.

## Impaired cholesterol absorption exacerbates synaptic deficits under low-cholesterol conditions and hinders the increase in synapses under high-cholesterol conditions

Because *ncr-1(0)* mutants exhibited lower levels of cholesterol, we proceeded to investigate whether cholesterol plays a role in the regulation of synapse development mediated by NCR-1. We grew WT animals expressing SNB-1::GFP/VAMP in GABAergic or cholinergic (*unc-129* promoter) motor neurons in low (0 μg/ml), normal (10 μg/ml), and high (40 μg/ml) dietary cholesterol. We analyzed presynaptic boutons in the dorsal cord (Fig 1E). Because autoclaving cholesterol did not affect synapse number, we added cholesterol to nematode growth medium preparations before autoclaving. When

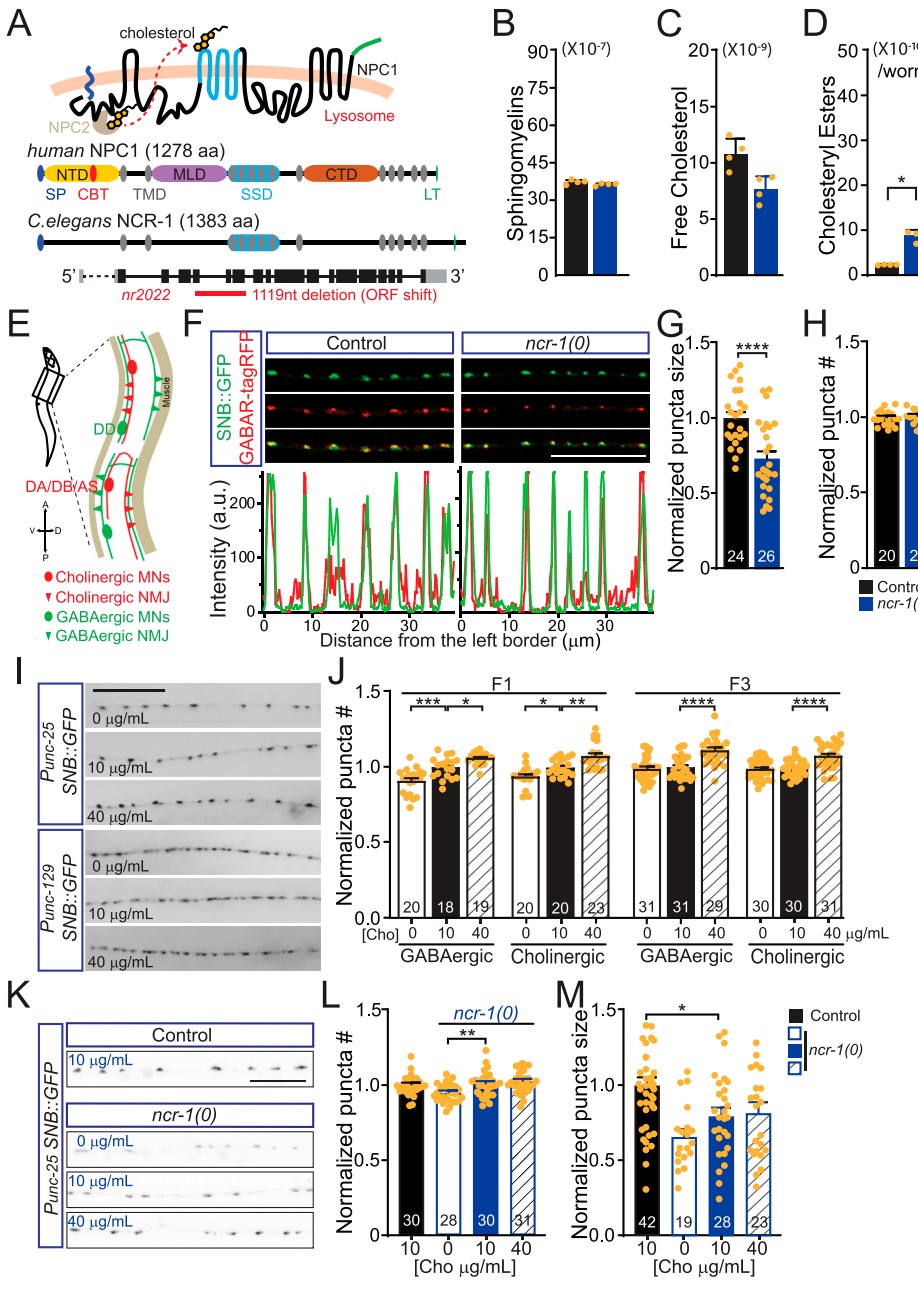

**Figure 1. Loss of function of NCR-1 impairs cholesterol absorption and leads to synaptic deficits.**

**(A)** Schematic illustration of the structures of human NPC1 and *C. elegans* NCR-1, as well as a summary of the gene structure and null allele of *ncr-1*. NPC1 accepts unesterified cholesterol from soluble luminal NPC2 and passes cholesterol from its N-terminal domain to the transmembrane sterol-sensing domain, allowing cholesterol inserted into the lysosomal membrane before egress. Human NPC1 comprises 3 large luminal domains and 13 transmembrane passes. The corresponding domains in these proteins are in the same shape and color. Abbreviations for protein domains are as follows: SP, signal peptide; NTD, N-terminal domain; CBT, cholesterol binding and transfer domain; TMD, transmembrane domain; MLD, middle domain; SSD, sterol-sensing domain; CTD, C-terminal domain; LT, lysosome-targeting motif. The *nr2022* allele causes open reading frame shift and is a null allele of *ncr-1*. **(B, C, D)** Abundance of sphingomyelins, free cholesterol, and cholesteryl esters in control and *ncr-1(0)* mutant worms. Shown are average abundance per worm ± SD. **(B, C, D)** Statistics, Kruskal–Wallis test and uncorrected Dunn's multiple comparison post-test: (B) $P$ = 0.2347; (C) $P$ = 0.1813; and (D) *$P$ < 0.05, $P$ = 0.02589. n = 4 biologically independent samples. The source data are provided as a separate file (Excel S1). **(E)** Illustration of presynaptic boutons in the dorsal cord of *C. elegans*, primarily from DD-type GABAergic and DA/DB/AS-type cholinergic motor neurons. **(F)** Presynaptic SNB-1::GFP is opposed by postsynaptic GABAR-tagRFP in *ncr-1(0)* mutants. Images show synapses of the GABAergic motor neurons labeled by *juIsI [Punc-25-SNB::GFP]* and *KrSi2 [Punc-49::unc-49B-tagRFP]* in control and *ncr-1(0)* mutants. Scar bar: 20 μm. **(G, H)** Quantification of the size and the number of synapses in genotypes indicated. The average number of control animals was normalized per individual. Shown are average values ± SEM (mean ± SEM). The numbers of animals analyzed are indicated. Statistics, unpaired *t* test (two-tailed): ****$P$ < 0.0001 (size). **(I)** Representative images showing synapses of the GABAergic and cholinergic motor neurons labeled by *juIsI[Punc-25-SNB::GFP]* or by *nuIs152[Punc-125-SNB::GFP]* under different cholesterol conditions after at least three

generations. Scar bar: 20 μm. **(J)** Quantitative analysis of the number of GABAergic and cholinergic synapses under different cholesterol conditions. L4 worms fed with low (0 μg/ml), normal (10 μg/ml), or high (40 μg/ml) dietary cholesterol were transferred to plates with the same cholesterol concentration for one or three generations. The average synaptic number of control animals was normalized per individual. Shown are average values ± SEM (mean ± SEM). The numbers of animals analyzed are indicated. Statistics, Kruskal–Wallis test and Dunn's multiple comparison post-test or one-way ANOVA and Bonferroni's multiple comparison post-test: *$P$ < 0.05, $P$ = 0.0142 (GABAergic), $P$ = 0.0420 (cholinergic); **$P$ < 0.01, $P$ = 0.0096; ***$P$ < 0.001, $P$ = 0.0004; ****$P$ < 0.0001. **(K)** Representative images showing synapses of the GABAergic motor neurons in genotypes indicated under different cholesterol conditions. L4 worms grown in different concentrations of cholesterol were transferred to plates with the same cholesterol concentration for >3 generations. Scar bar: 10 μm. **(L, M)** Quantitative analysis of the number and size of GABAergic synapses in genotypes indicated under different cholesterol conditions. The average number of control animals (fed with 10 μg/ml cholesterol) was normalized per individual. Shown are average values ± SEM (mean ± SEM). The numbers of animals analyzed are indicated. Statistics, Kruskal–Wallis test and Dunn's multiple comparison post-test (puncta #) or one-way ANOVA and Bonferroni's multiple comparison post-test (punctate size): *$P$ < 0.05, $P$ = 0.0189 (size); **$P$ < 0.01, $P$ = 0.0084 (number).
Source data are available for this figure.

grown in a low-cholesterol diet for more than three generations, WT animals exhibited a similar number of synapses in both low- and normal-cholesterol conditions (Fig 1I and J). Interestingly, the F1

generation of WT animals had fewer synapses compared with those grown in normal dietary cholesterol (Fig 1J). Extended exposure to low cholesterol did not appear to impair the formation of synapses,

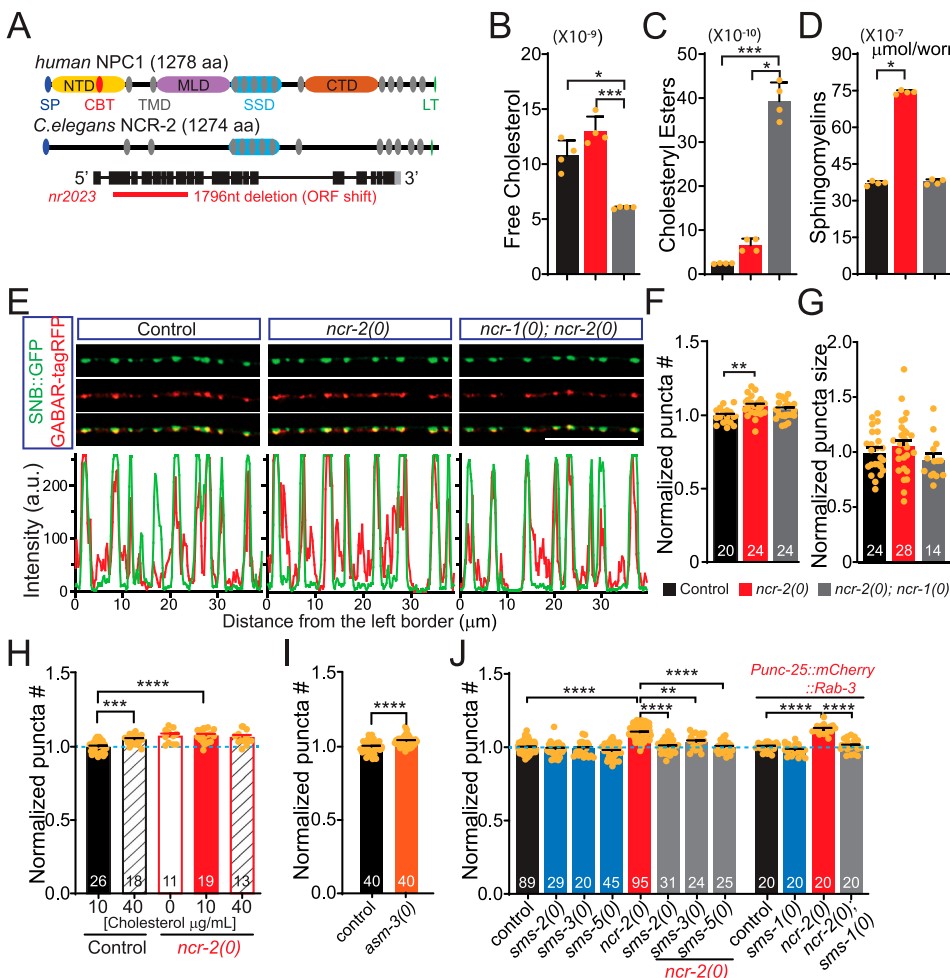

**Figure 2. Loss of function of NCR-2 impairs the use of cholesterol and sphingomyelins and promotes synapse formation.**
**(A)** Schematic illustration of the structures of human NPC1 and *C. elegans* NCR-2, as well as a summary of the gene structure and null allele of *ncr-2*. The *nr2023* allele causes open reading frame shift and is a null allele of *ncr-2*. **(B, C, D)** Abundance of free cholesterol, cholesteryl esters, and sphingomyelins in control, *ncr-2(0)*, and *ncr-1(0)*; *ncr-2(0)* mutant worms. Shown are average abundance per worm ± SD. **(B, C, D)** Statistics, Kruskal–Wallis test and uncorrected Dunn's multiple comparison post-test: (B) *$P < 0.05$, $P = 0.01157$; ***$P < 0.001$, $P = 0.000635$; $P = 0.02589$ (*ncr-2(0)* versus *ncr-1(0)*); (C) *$P < 0.05$, $P = 0.02589$; ***$P < 0.001$, $P = 0.000364$; and (D) *$P < 0.05$, $P = 0.0313$; $P = 0.0008$ (*ncr-2(0)* versus *ncr-1(0)*). n = 4 biologically independent samples. The source data are provided as a separate file (Excel S1). **(E)** Presynaptic SNB-1::GFP is opposed by postsynaptic GABAR-tagRFP in control, *ncr-2(0)*, and *ncr-1(0)*; *ncr-2(0)* mutants. Images show synapses of the GABAergic motor neurons labeled by *juIs1 [Punc-25-SNB::GFP]* and *KrSi2[Punc-49::unc-49B-tagRFP]*. Scar bar: 20 μm. **(F, G)** Quantification of the number and size of synapses in genotypes indicated. The average number of control animals was normalized per individual. Shown are average values ± SEM (mean ± SEM). The numbers of animals analyzed are indicated. Statistics, one-way ANOVA and Bonferroni's multiple comparison post-test: **$P < 0.01$, $P = 0.0014$. **(H)** Quantitative analysis of the number of GABAergic synapses in genotypes indicated under different cholesterol conditions. L4 worms grown in different concentrations of cholesterol were transferred to plates with the same cholesterol concentration for more than three generations. The average number of control animals (fed with 10 μg/ml cholesterol) was normalized per individual. The numbers of animals analyzed are indicated. Statistics, one-way ANOVA and Bonferroni's multiple comparison post-test: ***$P < 0.001$, $P = 0.0006$; ****$P < 0.0001$. **(I)** Quantitative analysis of the number of GABAergic synapses in genotypes indicated. The average number of control animals was normalized per individual. *asm-3(0)* mutants show increased synapses. The numbers of animals analyzed are indicated. Statistics, unpaired *t* test: ****$P < 0.0001$. **(J)** Quantitative analysis of the number of GABAergic synapses in genotypes indicated. The average number of control animals was normalized per individual. The numbers of animals analyzed are indicated. Statistics, Kruskal–Wallis test and Dunn's multiple comparison post-test (*juIs1* group) or one-way ANOVA and Bonferroni's multiple comparison post-test (*juIs236* group): ****$P < 0.0001$; **$P < 0.01$, $P = 0.0012$.
Source data are available for this figure.

presumably because of residual cholesterol in the agar medium and the minimal cholesterol requirement for synapse development (48, 49). In high dietary cholesterol, the synapse number of WT animals increased nearly 10% (Fig 1I and J). Therefore, excessive dietary cholesterol promotes the development of synapses in both GABAergic and cholinergic neurons.

Notably, *ncr-1(0)* mutants displayed an impaired increase in the number of synapses in high cholesterol (Fig 1K and L), suggesting that NCR-1–mediated cholesterol absorption is required for the formation of more synapses under high-cholesterol conditions. When grown in low cholesterol, *ncr-1(0)* mutants exhibited even more severe synaptic deficits, as evidenced by a more pronounced reduction in both the number and size of synapses compared with their counterparts grown in normal cholesterol (Fig 1L and M). These results suggested that impaired cholesterol absorption contributes to the synaptic deficits in *ncr-1(0)* mutants.

## NCR-2 promotes the use of cholesterol and sphingomyelins and impedes synapse development

Given the potential of NCR-1 to elevate cholesterol levels and promote synapse development, it is pertinent to inquire about the factors that exert the opposite effects. NCR-2, another cholesterol transporter in *C. elegans*, also shares the same structure domains as NPC1 (40), with 27% identity and 68% similarity to the human NPC1 protein (Figs 2A and S2). Null mutants of *ncr-2*, resulting from the *nr2023* deletion allele (43) (Fig 2A), exhibited no significant alterations in the overall lipid levels in the 10 subcategories (TAGs, DAGs, FFAs, PCs, PAs, PSs, PIs, PEs, CLs, and PGs) (Fig S3C–L). Interestingly, the *ncr-2(0)* mutants displayed higher levels of free cholesterol compared with the WT, and this increase was particularly pronounced when compared to the *ncr-1(0)* mutants (Fig 2B and Supplemental Source data file). Similar to *ncr-1(0)* mutants, *ncr-2(0)* mutants showed higher levels of cholesteryl ester

compared with the WT (Fig 2C), supporting that NCR-2 also aids in the hydrolysis or inhibits the formation of cholesteryl esters. Together, these observations support the role of NCR-2 in cholesterol use, distinct from NCR-1's role in cholesterol absorption. Furthermore, a significant increase in sphingomyelin levels was observed in the ncr-2(0) mutants compared with both the WT and ncr-1(0) mutants (Fig 2D), suggesting that NCR-2 also facilitates the use of sphingomyelins.

In contrast to the phenotypes observed in ncr-1(0) mutants, ncr-2(0) mutants showed a 7% increase in GABAergic presynaptic boutons of normal size compared with the WT (Fig 2E–G). Moreover, all the presynaptic boutons in ncr-2(0) mutants were also accompanied by ionotropic GABA receptors (Fig 2E). Therefore, unlike NCR-1, NCR-2, the other C. elegans homolog of human NPC1, facilitated the use of cholesterol and sphingomyelin and hindered synapse development. Instead of resembling the synaptic phenotypes in either ncr-1(0) or ncr-2(0) single-mutant counterparts, ncr-1(0); ncr-2(0) double mutants showed comparable number and size of synapses to WT animals (Fig 2E–G), supporting that NCR-1 and NCR-2 are involved in distinct parallel pathways.

### Reduced use of cholesterol and sphingomyelins is sufficient to promote synapse formation

As ncr-2(0) mutants exhibited higher levels of cholesterol and sphingomyelin in comparison with WT, we next explored whether the synaptic phenotypes observed in ncr-2(0) mutants were attributable to the elevated cholesterol and sphingomyelin levels. The ncr-2(0) mutants showed an equivalent increase in synapses under conditions of low, normal, and high cholesterol (Fig 2H), suggesting that the decreased use of cholesterol is sufficient to induce more synapses. The synthesis and degradation of sphingomyelins are facilitated by enzymes known as sphingomyelin synthase and sphingomyelinases (50). Interestingly, the loss of sphingomyelinase ASM-3, caused by the ok1744 deletion allele, results in a 4% increase in synapses similar to those found in ncr-2(0) mutants, albeit to a lesser extent (Fig 2I). These results suggest that the accumulation of sphingomyelins is also sufficient to induce more synapses. Consistent with this, the deletion alleles of multiple genes (sms-1, sms-2, sms-3, and sms-5) encoding sphingomyelin synthases, which are thought to disrupt sphingomyelin synthesis, completely suppressed the increase in synapses in ncr-2(0) mutants without causing any noticeable synaptic defects by itself (see the Materials and Methods section; Fig 2J). Although sphingomyelin levels have been reported highly correlated with cholesterol levels (51, 52, 53), these findings indicate that elevated levels of cholesterol and sphingomyelins are responsible for the synaptic phenotypes in ncr-2(0) mutants.

### NCR-1 and NCR-2 synergistically enhance the hydrolysis of cholesteryl esters

Mass spectrometry revealed alterations in cholesteryl ester levels in ncr-1(0) and ncr-2(0) mutants (Figs 1D, 2C, and S3). These findings promoted us to investigate whether lipid droplets, which store neutral lipids including cholesteryl esters, were affected. To label lipid droplets in vivo, we used transgenes carrying mCherry-fused

MDT-28, a ubiquitously expressed resident protein in lipid droplets (54). As predicted, mutations in daf-36, a key oxygenase converting cholesterol to 7-dehydrocholesterol (55), dramatically increased droplet size, whereas mutations in mboa-1, the homolog of mammalian acyl-coenzyme A: cholesterol acyltransferases (ACATs) catalyzing the cholesterol esterification (56), decreased droplet size (Fig 3A and B). Neither ncr-1(0) nor ncr-2(0) alone altered the morphology of lipid droplets; however, in ncr-1(0); ncr-2(0) double mutants, the size of lipid droplets increased significantly (Fig 3A and B). It is possible that the slight increase in cholesteryl ester levels caused by ncr-1(0) or ncr-2(0) alone is insufficient to induce morphological changes in lipid droplet. Interestingly, mboa-1(0) did not rescue the droplet size in ncr-1(0); ncr-2(0) mutants, suggesting that the increased droplet size is not due to elevated ACAT activity.

We further stained neutral lipids using Oil Red O (ORO), a lipophilic dye validated to accurately assess major fat storage in worms. Consistent with previous studies (57), daf-2 mutants exhibited a significant increase in fat storage (Fig 3C and D). In line with lipid droplet data, we observed a significant increase in fat accumulation exclusively in ncr-1(0); ncr-2(0) mutants. Although mboa-1(0) alone decreased body fat stores, it did not prevent the increase of body fat levels in ncr-1(0); ncr-2(0) mutants (Fig 3E). Together, these findings suggest that instead of inhibiting cholesterol esterification, NCR-1 and NCR-2 collaborate to promote the hydrolysis of cholesteryl esters.

### NCR-1 and NCR-2 exhibit distinct effects on cholesterol distribution

Both NCR-1 and NCR-2 promote cholesterol mobilization. To investigate subcellular distribution of free cholesterol in N2 and mutant animals, we used filipin, a naturally fluorescent polyene antibiotic that specifically binds non-esterified sterols. By analyzing the pharynx of filipin-stained animals, we found that ncr-2(0) mutants exhibited increased cholesterol levels in the pharynx compared with N2 animals, whereas ncr-1(0) and ncr-1(0); ncr-2(0) mutants showed decreased cholesterol levels (Fig 3F and G). When animals were maintained on a low-cholesterol diet, the cholesterol levels in both N2 and ncr-2(0) mutants decreased significantly (Fig 3G), implying that dietary cholesterol can influence the worm's cholesterol content. However, we did not observe evident cholesterol distribution in the nerve ring, a structure rich in synapses near the pharynx, suggesting that the amount of cholesterol required for neuronal development is low.

We further observed the distribution of 25-NBD, a fluorescent analog of cholesterol, by feeding it to worms. We detected clear signals in the embryos of adult animals, rather than in somatic cells. In contrast to the filipin results, ncr-2(0) mutants exhibited decreased cholesterol levels in the embryos, whereas ncr-1(0) and ncr-1(0); ncr-2(0) mutants displayed increased cholesterol levels compared with N2 animals (Fig S4A and B). Intriguingly, the ncr-1(0) mutants grown in low cholesterol showed further increased cholesterol levels in the embryos (Fig S4A and B). These results suggest that the uptake of cholesterol in the embryos is mediated by an independent pathway not involving NCR-1 and NCR-2.

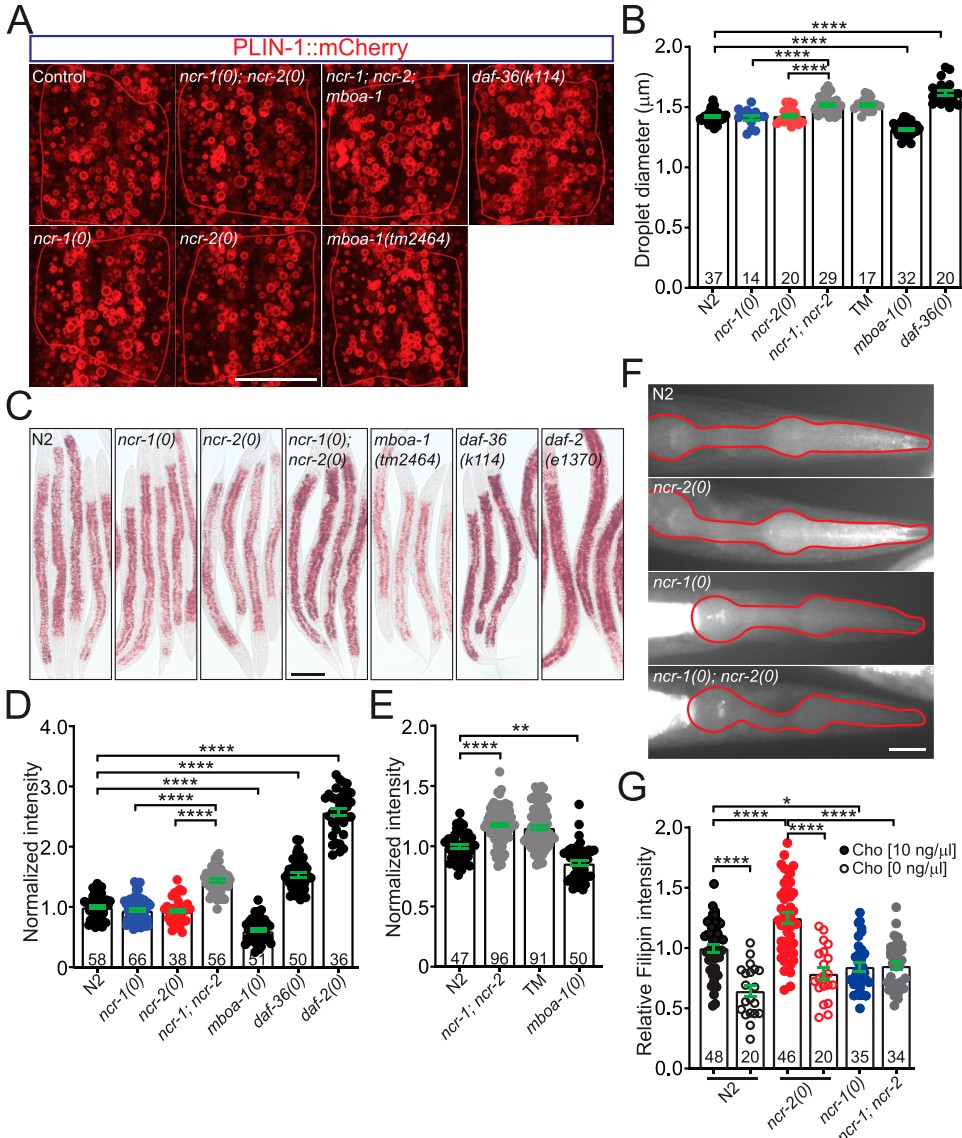

**Figure 3. NCR-1 and NCR-2 synergistically enhance the hydrolysis of cholesteryl esters and have distinct effects on cholesterol distribution.**
**(A)** Representative images of lipid droplets (LDs), visualized using PLIN-1::mCherry, in middle L4 worms of genotypes indicated. Shown are the LDs in the int2 left and right cells. The cell boundary of int2 cells is highlighted in red. Scale bar: 20 $\mu$m. **(B)** Quantification of droplet size in genotypes indicated. Each dot represents the average diameter of lipid droplets in int2 cells of a middle L4 worm. The numbers of animals analyzed are indicated. TM refers to the triple mutants *ncr-1(0)*; *ncr-2(0)*; *mboa-1(0)*. Statistics, one-way ANOVA and Bonferroni's multiple comparison post-test: ****$P$ < 0.0001. **(C)** Oil Red O staining in middle L4 worms of genotypes indicated. At least three independent experiments were performed with similar results. Scale bar: 1 mm. **(D)** Quantification of Oil Red O signals in (C). The average total intensity of WT animals was normalized per individual for each experiment. Each dot represents the normalized values of a worm. Bars show average values ± SEM (mean ± SEM). The numbers of animals analyzed are indicated. Statistics, Kruskal–Wallis test and Dunn's multiple comparison post-test: ****$P$ < 0.0001. **(E)** Increase in body fat storage in *ncr-1(0)*; *ncr-2(0)* double mutants is not dependent on *mboa-1*. The graph illustrates the quantification of ORO signals in control, *mboa-1(tm2464)* single mutants, *ncr-1(0)*; *ncr-2(0)* double mutants, and *ncr-1(0)*; *ncr-2(0)*; *mboa-1(2464)* triple mutants. The average total intensity of WT animals was normalized per individual for each experiment. Each dot represents the normalized values of a worm. Bars show average values ± SEM (mean ± SEM). The numbers of animals analyzed are indicated. Statistics, Kruskal–Wallis test and Dunn's multiple comparison post-test: **$P$ < 0.01, $P$ = 0.0063; ****$P$ < 0.0001. **(F)** NCR-1 and NCR-2 exhibit differential effects on cholesterol distribution. Representative images of filipin-stained pharynx in L4 animals grown under normal-cholesterol conditions (10 ng/$\mu$l) are shown. Worms of various genotypes were maintained in either low (0 ng/$\mu$l) or normal cholesterol (10 ng/$\mu$l) for more than four generations. When grown in normal cholesterol, the *ncr-2(0)* mutant animals displayed higher filipin fluorescence in the pharynx, whereas *ncr-1(0)*; *ncr-2(0)* mutants displayed lower filipin fluorescence in the pharynx compared with N2 animals. **(G)** Quantification of filipin intensity in the pharynx of animals of genotypes indicated. The pharynx of each L4 worm was delineated, and the mean intensity was measured. The average value of N2 animals (fed with 10 $\mu$g/ml cholesterol) was normalized per individual. Each dot represents the normalized value of a single animal, with bars indicating average values ± SEM (mean ± SEM). The numbers of animals analyzed are indicated. Statistics, one-way ANOVA and Bonferroni's multiple comparison post-test: ****$P$ < 0.0001; *$P$ < 0.05, $P$ = 0.022.

## Sterol genes potentially associated with NCR-1 or NCR-2 pathways

We next investigate the potential effects of *ncr-1(0)* or *ncr-2(0)* on neuronal morphology. To visualize the morphology of GABAergic motor neurons, we used transgenic animals expressing *unc-25p::GFP*. We observed grossly normal morphology of neurons and axons in all mutants (Fig 4A). The proportion of animals and the proportion of neurons with ectopic branches remained unaltered by either *ncr-1(0)* or *ncr-2(0)* (Fig 4B and C), suggesting that NCR-1 and NCR-2 are specifically involved in regulating synapse formation.

We further performed a cherry-picked genetic screen to identify additional mutations in cholesterol metabolism genes that influence synapses under a high-cholesterol diet. We found that mutations in the nuclear hormone receptors NHR-80 and NHR-8 did not impede the increase of synapses in high cholesterol (Fig 4D), implying that they are not involved in the NCR-1 or NCR-2 pathways. In contrast, mutations in VIT-1 and VIT-2, which are two major apolipoproteins for cholesterol trafficking between tissues, and in HEH-1, an ortholog of human NPC2, which is a soluble luminal protein in lysosomes and responsible for transferring cholesterol to human NPC1 (39, 58), hindered the effects of high cholesterol as

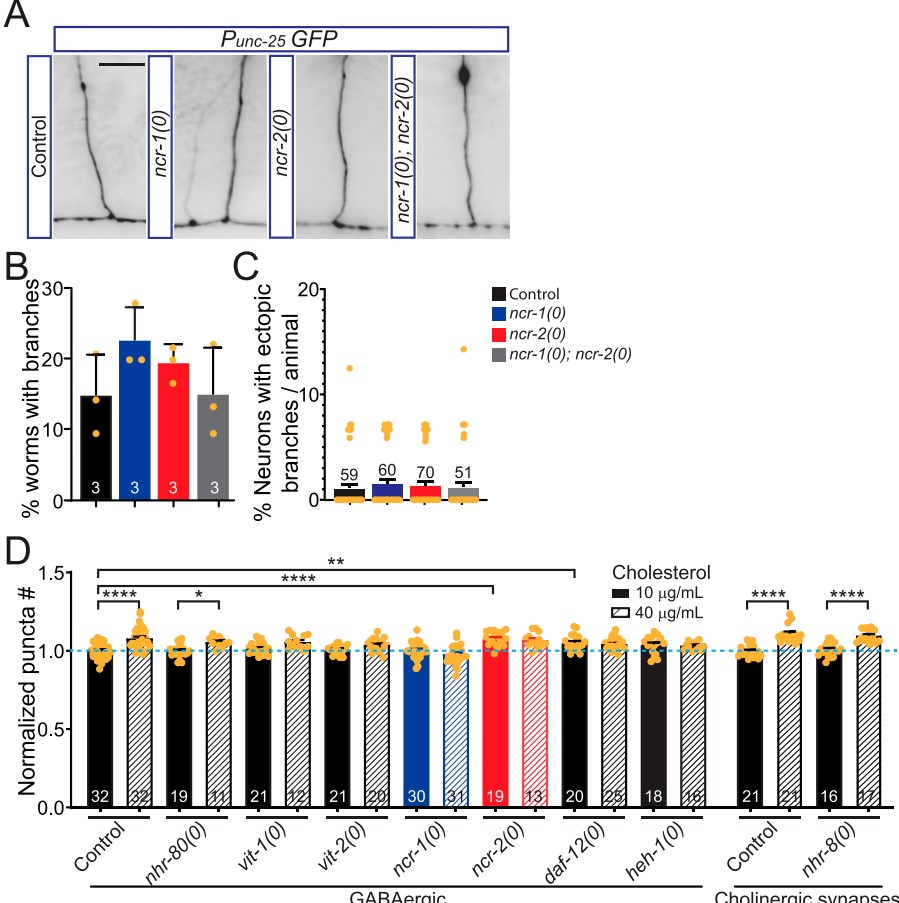

**Figure 4. Sterol genes that are potentially associated with the specific role of NCR-1 and NCR-2 in synapse development.**
**(A)** Representative images showing axonal morphology of the GABAergic motor neurons labeled by *juIs76 [Punc-25-GFP]* in genotypes indicated. In *ncr-1(0)*, *ncr-2(0)*, and *ncr-1(0); ncr-2(0)* mutants, commissures show similar morphology as controls. Scale bar: 10 *μm*. **(B)** Quantitation of the percentage of animals with ectopic branches in genotypes indicated. Three independent experiments were performed. In each experiment, 14–25 1-d adults of each genotype were scored, and 14–18 commissures were counted per animal. Data are shown as the average percentage of animals with ectopic branches ± SD (n = 3 experiments). **(C)** Quantitation of the percentage of neurons with ectopic branches per animal in genotypes indicated. Data are shown as the average percentages of abnormal neurons per animal ± SEM (n = number of animals). **(D)** Screened subset of cholesterol genes for synaptic phenotypes labeled by *juIs1 [Punc-25-SNB::GFP]* or *nuIs152 [Punc-129-SNB::GFP]*. Total synaptic puncta in the dorsal nerve cord were counted, and the average number of controls was normalized per individual. The diagram shows average values ± SEM (mean ± SEM). The numbers of animals analyzed are indicated. Statistics, one-way ANOVA and Bonferroni's multiple comparison post-test (GABAergic synapses) or Kruskal–Wallis test and Dunn's multiple comparison post-test (cholinergic synapses): *P < 0.05, P = 0.0244; **P < 0.01, P = 0.0021; ****P < 0.0001.

*ncr-1(0)* (Fig 4D). These results suggest that NCR-1 functions analogously to NPC1 and that cholesterol trafficking and uptake are essential for the synaptic increase observed in high-cholesterol conditions. Mutations in DAF-12, a nuclear receptor regulated by cholesterol-derived dafachronic acids that control reproductive growth, resulted in increased synapse formation even under normal-cholesterol conditions (Fig 4D). This phenotype is similar to that observed in *ncr-2(0)* mutants, suggesting a potential relevance of DAF-12 to NCR-2 pathways.

## NCR-1 and NCR-2 both act in neurons to influence synapse development

We next addressed the cell-type requirement of *ncr-1* and *ncr-2* in synaptic regulation. *ncr-1(0); ncr-2(0)* mutants usually form dauer larvae and have an extremely small brood size in low cholesterol (41) (Fig S5A). The transgenic expression of the genomic coding sequences for *ncr-1* or *ncr-2*, under the control of the broadly expressed *sur-5* promoter or the XXX cell-specific *eak-4* promoter, rescued the brood size of *ncr-1(0); ncr-2(0)* mutants (Fig S5B–D). These results support that the cloned genomic sequences of *ncr-1* and *ncr-2* encode functional NCR-1 and NCR-2 proteins.

The transgenic expression of NCR-1, either broadly or specifically in neurons, was sufficient to fully restore synaptic size in *ncr-1(0)*

mutants. In contrast, NCR-1 expression in muscle did not yield the same effects (Fig 5A–C). As a control, the overexpression of NCR-1, driven by different promoters, did not produce any notable synaptic changes in WT animals (Fig 5A–C). These findings suggest that NCR-1 functions in neurons to promote synapse development. To further investigate in which tissue NCR-2 functions, we expressed NCR-2 in specific tissues in *ncr-1(0); ncr-2(0)* mutants and analyzed synaptic size. We found that the transgenic expression of *ncr-2(+)* under the control of the *sur-5* promoter reduced the synaptic size of *ncr-1(0); ncr-2(0)* to levels comparable to *ncr-1(0)* mutants (Fig 5D–F). In addition, the expression of NCR-2 in either neurons or muscles also significantly decreased synaptic size in *ncr-1(0); ncr-2(0)* mutants (Fig 5D–F). The broad (*Sur-5p*), neuronal (*F25B3.3p*), or muscular (*myo-3p*) expression of NCR-2 did not yield any detectable changes in the synapses of WT animals (Fig 5D–F). These findings suggest that although NCR-2 may have non–neuron-autonomous roles in regulating synapses, both NCR-1 and NCR-2 can influence synapse development in neurons.

## Endogenous NCR-1 and NCR-2 respond to low-cholesterol feeding

Previous studies investigating NCR-1 and NCR-2 expression used fluorescence reporters driven by various hypothetical promoters. Various lengths of *ncr-1* upstream promoter sequences induce

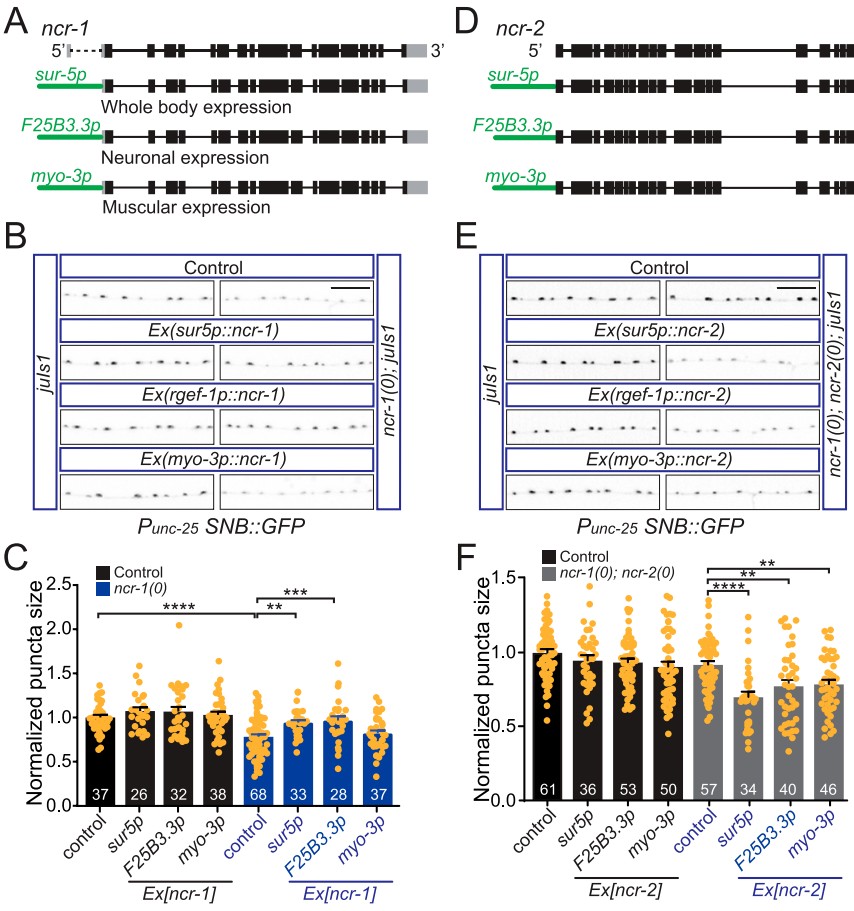

**Figure 5. NCR-1 and NCR-2 act in neurons to influence synapse development.**
**(A)** Gene structure of the *ncr-1* loci and expression constructs. Genomic sequences of *ncr-1* starting from 5'UTR to 3'UTR were cloned into vectors containing different promoters. Black boxes represent coding exon sequences, and gray boxes represent untranslated regions. **(B)** Expression of NCR-1 in neurons rescues synaptic deficits in *ncr-1(0)*. Images show synaptic puncta in the dorsal nerve cord, posterior to the vulva region, of the GABAergic motor neurons labeled by *juIs1*. *ncr-1(0)* mutants show smaller synaptic puncta compared with control animals. The whole-body expression (*sur-5p*) or neuronal expression (*F25B3.3p*) of NCR-1 rescued synaptic defects in *ncr-1(0)*, whereas muscular expression did not. The tissue-specific expression of NCR-1 in *juIs1* did not alter the synaptic morphology. Scale bar: 10 μm. **(C)** Quantification of synaptic size in strains with the tissue-specific expression of NCR-1. Confocal Z-stack images of the nerve cords (0.5 μm/section) were collected for quantitative analysis. The average area of *juIs1* was normalized per individual in the same experiment. Shown are average values ± SEM. The numbers of animals analyzed are indicated. Statistics, one-way ANOVA and Bonferroni's multiple comparison post-test: ****P < 0.0001; **P < 0.01, P = 0.0023; ***P < 0.001, P = 0.0007. **(D)** Gene structure of the *ncr-2* loci and expression constructs. Genomic sequences of *ncr-2* starting from the start codon to 3'UTR were cloned into vectors containing different promoters. **(E)** Expression of NCR-2 in neurons or muscle cells rescues the suppression of *ncr-1(0)* by *ncr-2(0)*. Images show synaptic puncta in the dorsal nerve cord, posterior to the vulva region, of the GABAergic motor neurons labeled by *juIs1*. *ncr-1(0)*; *ncr-2(0)* mutants show similar synaptic puncta as control animals. The whole-body expression (*sur-5p*), neuronal expression (*F25B3.3p*), or muscular expression (*myo-3p*) of NCR-2 rescued the suppression of *ncr-1(0)* by *ncr-2(0)*. The neuronal or muscular expression of NCR-2 in *juIs1* did not notably alter the synaptic morphology. Scale bar: 10 μm. **(F)** Quantification of the synaptic size in the tissue-specific expression of NCR-2 strains. The average area of *juIs1* was normalized per individual in the same experiment. Shown are average values ± SEM. The numbers of animals analyzed are indicated. Statistics, one-way ANOVA and Bonferroni's multiple comparison post-test: ****P < 0.0001; **P < 0.01, P = 0.0043 (*F25B3.3p*), P = 0.0077 (*myo-3p*).

transcriptional activity in multiple tissues, including head and tail neurons. The *ncr-2* upstream promoter sequences demonstrate transcriptional activity in XXX cells, and in the gonadal sheath and the ventral nerve cord (41, 59).

To investigate the endogenous expression pattern of NCR-1 and NCR-2, we used CRISPR/Cas9 genome editing to tag the endogenous *ncr-1* and *ncr-2* genes (see the Materials and Methods section). We inserted *gfp::3Xflag* into the cytoplasmic tail of NCR-1 and NCR-2 (Fig S6A and B, *cfu39(ncr-1::GFP)* and *cfu45(ncr-2::GFP)*; Supplemental Methods). The *ncr-1(cfu39)* or *ncr-2(cfu45)* knock-in alleles did not disrupt NCR-1 or NCR-2 function, as evidenced by the observation that *ncr-1(cfu39)*; *ncr-2(0)* or *ncr-2(cfu45)*; *ncr-1(0)* animals had similar brood size compared with their *ncr-2(0)* or *ncr-1(0)* single-mutant counterparts in low cholesterol (Fig S6C).

We then followed the stage-specific patterns of NCR-1 and NCR-2. At larval stages, NCR-1::GFP(*cfu39*) was observed at the peripheries of intestine and epidermal seam cells (Fig S6D, upper panels). In adult hermaphrodites, NCR-1::GFP was evident in the proximal gonadal sheath cells and spermatheca (Fig S6D, upper panels). We did not detect significant levels of GFP fluorescence in the head and ventral nerve cord at various stages of development

(Fig S6D, upper panels), probably reflecting the low expression of NCR-1 in these regions. It is worth noting that low dietary cholesterol feeding resulted in a significant increase in NCR-1::GFP expression, specifically in the head region including the pharynx, XXX cells, neurons, and sheath cells (Fig S6D and F). This increase was not influenced by *ncr-2(0)* (Fig S6F), suggesting that cholesterol inhibits the endogenous head expression of NCR-1 independently of NCR-2.

Unlike NCR-1::GFP, NCR-2::GFP(*cfu45*) was predominantly observed in the cytoplasm and periphery of sperm cells; this expression in sperm cells might be further increased by low-cholesterol feeding (Fig S6E, upper and lower panels). Moreover, NCR-2::GFP was observed in two XXX cells of 23% of normally fed L1-L3 animals, with 55% of animals showing NCR2::GFP expression in one XXX cell. Interestingly, low-cholesterol feeding increased the percentage to 49%, with 33.5% of animals exhibiting NCR-2::GFP expression in one XXX cell (Fig S6G). The *ncr-1(0)* mutation did not affect this outcome (Fig S6G). These results imply that cholesterol availability affects NCR-2 signaling in XXX cells independently of NCR-1. Taken together, these CRISPR knock-in lines revealed distinct expression patterns of the endogenous NCR-1 and NCR-2. Under low cholesterol, the expression of these two NPC1

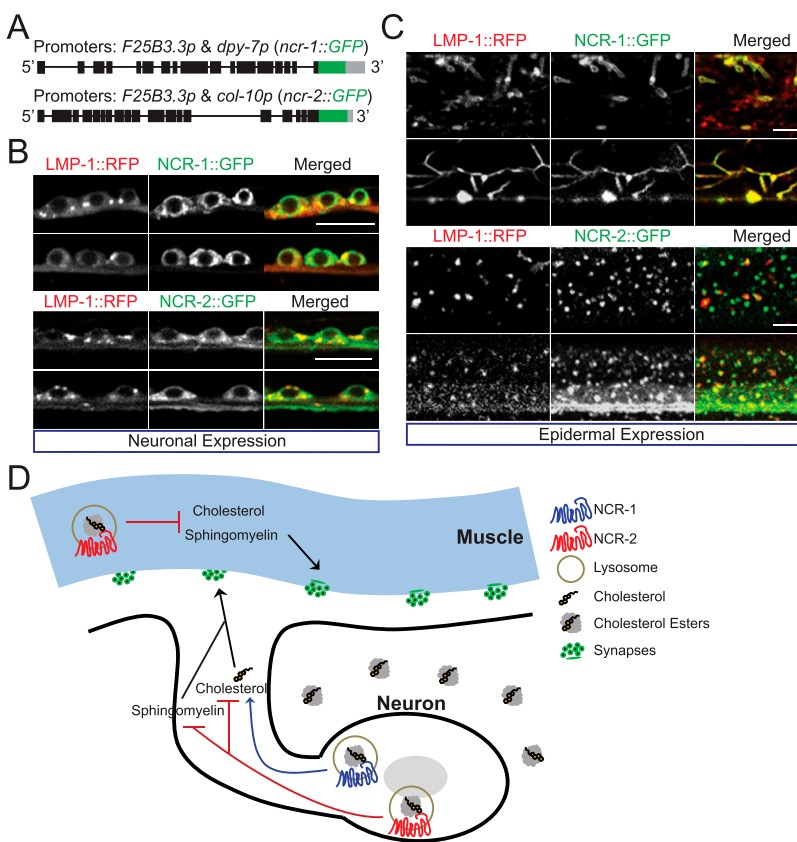

**Figure 6. NCR-1 and NCR-2 are lysosome-associated proteins.**
**(A)** Schematic illustration for constructs that express NCR-1::GFP and NCR-2::GFP in neurons or epidermis. **(B)** In the transgenic lines expressing both the lysosomal marker *unc-119p::lmp-1::RFP* and either *F25B3.3p::ncr-1::GFP* or *F25B3.3p::ncr-2::GFP*, GFP and RFP signals partially overlap in motor neurons along the ventral nerve cord. For each condition, 1-d adults of 2–3 lines were imaged. Scale bar: 10 μm. **(C)** In the transgenic lines expressing both the lysosomal marker *col-12p::lmp-1::RFP* and either *dpy-7p::ncr-1::GFP* or *col-10p::ncr-2::GFP*, GFP and RFP signals overlap in the epidermal cells. For each condition, L4 worms of 2–3 lines were imaged. Scale bar: 10 μm. **(D)** Cartoon illustrating the working model. Lysosomal cholesterol transporters, NCR-1 and NCR-2, have distinct functions in controlling cholesterol homeostasis, consequently exerting a significant impact on synaptic development. In particular, NCR-1 promotes synapse formation by increasing cholesterol absorption, whereas NCR-2 inhibits synapse formation by using cholesterol and sphingomyelins.

homologs showed notable alterations, reinforcing their roles in regulating cholesterol homeostasis.

## NCR-1 and NCR-2 are lysosome-associated proteins

Human NPC1 contains a dileucine lysosome-targeting motif (LLNF) in the cytoplasmic tail (Figs S1 and S2) (60) and is located in a subset of the endosome/lysosome (61, 62). Based on protein sequences, NCR-1 has [D/E]XXXL[L/I]- and NCR-2 has YXXϴ-type (X, any amino acid; ϴ, an amino acid with a bulky hydrophobic side chain) lysosome-targeting signal in the cytoplasmic tail (63) (Figs S1 and S2).

Because of the low expression levels of NCR-1 and NCR-2 in multiple tissues (Fig S6D and E), we employed overexpression to determine their subcellular localization. Given that both NCR-1 and NCR-2 function in neurons (Fig 5), we used a pan-neuronal promoter (*F25B3.3p*) to overexpress NCR-1::GFP and NCR-2::GFP specifically in neurons (Fig 6A). Both GFP-tagged proteins are functional, as GFP was inserted after the same residues as *cfu39(ncr-1::GFP)* and *cfu45(ncr-2::GFP)* (Fig S6; Supplemental Methods). We also visualized lysosomes in neurons by overexpressing RFP-tagged LMP-1 under the control of another neuronal promoter (*unc-119p*). The *lmp-1* gene in *C. elegans* encodes the ortholog of lysosome-associated LAMP2 (64). LMP-1::RFP expression in neurons displayed a diffuse distribution, with sporadic formation of puncta. The neuronal co-expression of LMP-1::RFP with NCR-1::GFP or NCR-2::GFP resulted in partial co-localization of RFP and GFP signals (Fig 6B).

We further overexpressed NCR-1::GFP or NCR-2::GFP in the syncytium epidermal cells using epidermal promoters (*dpy-7p* or *col-10p*) (Fig 6A). Interestingly, the overexpression of either NCR-1::GFP or NCR-2::GFP in the epidermis resulted in lysosomes exhibiting primarily tubular structures or small puncta at the L4 stage (Fig 6C). Tubular lysosomes increase in abundance with age and exhibit reduced dynamism and activity (65). Although NCR-1 may have a role in modulating lysosomal activity and potentially influencing aging, NCR-1::GFP showed complete co-localization with LMP-1::RFP in the epidermal cells (Fig 6C, upper panels). In contrast to NCR-1::GFP, NCR-2::GFP exhibited small punctate structures similar to those of LMP-1::RFP in the epidermal cells at the L4 stage (Fig 6C, lower panels). Notably, there was a significant co-localization of RFP and GFP signals (Fig 6C, lower panels). These observations strongly support the notion that NCR-1 and NCR-2 are lysosome-associated proteins.

## Discussion

Using *C. elegans* motor neurons as a model, we have demonstrated the critical role of neuronal cholesterol homeostasis in synapse development. Our results reveal that two lysosomal cholesterol transporters have distinct functions in cholesterol homeostasis. In particular, NCR-1 promotes synapse development by increasing

cholesterol absorption, whereas NCR-2 hinders synapse development by facilitating the use of cholesterol and sphingomyelins (Fig 6D).

Both *ncr-1* and *ncr-2* have been implicated in intracellular cholesterol processing, as single or double mutants are unable to grow without cholesterol and their growth defects can be rescued by dafachronic acids derived from cholesterol (40, 41, 42). Our research demonstrates that NCR-1 and NCR-2 exert distinct effects on cholesterol levels. A prior study using dehydroergosterol (DHE), a natural fluorescent analog of cholesterol, did not reveal any changes in sterol distribution in worms fed bacteria containing dsRNA targeting the *ncr-1* genes (66). It is important to note that the outcomes of RNA interference and null mutants may differ. Furthermore, the esterified form of DHE exhibits fluorescence similar to non-esterified DHE, making them indistinguishable under a UV-sensitive wide-field microscope. Therefore, the accumulation of cholesteryl esters in worms treated with *ncr-1* dsRNA or lacking *ncr-1* might mask any reduction in cholesterol levels when employing the DHE method.

Although recent studies have shown that other lysosomal membrane proteins also facilitate cholesterol egress (67, 68, 69), the main source of knowledge regarding the role of lysosomal cholesterol transporters in the nervous system derives from research conducted on NPC1. NPC1 is expressed in both glia and neurons in the rodent brain (70, 71). The effects of NPC1 on the survival of Purkinje neurons (PNs) have been considered cell-autonomous. Deletion of *npc1* specifically in PNs causes PN loss in WT mice (72), whereas the expression of *npc1* specifically in PNs corrects cholesterol accumulation and prevents PN loss in *npc1−/−* mice (73). Interestingly, NPC1 also functions in a non–cell-autonomous manner. For example, the astrocyte-specific expression of NPC1 reduces neuronal cholesterol, delays neurodegeneration, and extends survival (74, 75, 76). Moreover, neuron-specific deletion of *npc1* blocks oligodendrocyte maturation and subsequent myelination (77), and cortical neurons co-cultured with NPC1-deficient astrocytes exhibit impaired neurite outgrowth (78). In the present study, NCR-1 influences synapse development in neurons, whereas NCR-2 impacts synapse development in both neurons and muscle cells (Fig 5). NCR-2 in muscles might hinder postsynaptic differentiation, which, in turn, compromises presynaptic differentiation. Alternatively, lipids or cholesterol-derived metabolites downstream of muscular NCR-2 could be excreted and transported across tissues to influence neurons. During hormone production and reproductive growth, NCR-2, mainly expressed in the endocrine XXX cells, synergizes with more broadly expressed NCR-1 to facilitate the production of cholesterol-derived dafachronic acids (DAs) (79). DAs, which are processed by DAF-9 (a cytochrome P450), propagate from XXX cells and stimulate more DA biosynthesis in the epidermis, thereby transactivating the vitamin D receptor DAF-12 throughout the animal and committing animals to reproductive development (79). Importantly, *ncr-1*; *ncr-2* have been positioned upstream of *daf-9* and *daf-12* in dauer pause regulation, as *daf-12(0)*, *daf-9* overexpression, sterol-derived DAF-9 products, and DAF-12 ligands all rescue dauer phenotypes of *ncr-1(0)*; *ncr-2(0)* mutants (41, 42). Our results show that *daf-12(0)* mutants phenotypically resemble *ncr-2(0)* mutants, suggesting that NCR-2 is also positioned upstream of DAF-12

to affect synapse development. Given that the neuronal or muscular expression of *ncr-2* hinders synapse formation but does not restore the brood size of *ncr-1(0)*; *ncr-2(0)* (Fig S5D), it is likely that DAF-12 ligands regulating synapse formation and reproduction differ. Consistent with this notion, some cholesterol-derived hormonal ligands for DAF-12 show varying activity in suppressing dauer formation and Mig or molting defects (42).

Previous studies have shed light on the functional significance of NPC1 in synapse development. In the primate cortex, NPC1 is primarily expressed in the astrocytic glial processes closely associated with terminals of axons and dendrites (80). In *npc1* null mice, the dendritic spine number of PNs is markedly reduced (81). In a mouse model with *npc1* mutation involving a single-nucleotide substitution, the morphology and function of hippocampal synapses are altered (82). Furthermore, the cerebellum in this model shows reduced GABAergic and glutamatergic synaptic inputs to PNs (83), impaired translocation of glutamatergic terminals from PN soma to proximal dendritic regions, and more glutamatergic inputs contacted or engulfed by microglia (84). However, it remains unclear whether these synaptic alterations caused by *npc1* deletion or mutation directly result from disturbed cholesterol homeostasis in neurons. Our findings demonstrate that lysosomal cholesterol transporters are crucial in controlling neuronal cholesterol homeostasis during synapse development. Both cholesterol and sphingomyelin are abundant in lipid rafts (12), and cholesterol traps sphingomyelin in nanoscale domains in the plasma membrane (51). Interestingly, cholesterol accumulates in human cells lacking sphingomyelinase or exposed to sphingomyelins (53, 85), and mutations in sphingomyelin synthase impair the absorption of exogenous cholesterol (52). Therefore, a strong correlation exists between cholesterol and sphingomyelin levels. Although sphingomyelin accumulation in our study may result from cholesterol buildup, reducing sphingomyelin levels completely suppresses the synaptic phenotypes. Our findings suggest potential therapeutic opportunities for targeting sphingomyelin levels to counteract NPC or other neurodegenerative diseases caused by disturbed cholesterol homeostasis.

Lysosomes act as a key sorting station for exogenous cholesterol. NPC1 binds to unesterified cholesterol within lysosomes and facilitates its transport to various cellular compartments through vesicular or non-vesicular trafficking mechanisms (86). In addition, lysosomal NPC1 is involved in signal transduction. For example, depletion of NPC1 in HEK293T- and iPSC-derived neurons results in abnormal cholesterol accumulation within lysosomes, leading to hyperactivation of mTORC1 kinase and mitochondrial dysfunction (87). Future studies are required to determine whether these two lysosomal cholesterol transporters influence diverse signaling pathways or cholesterol egress during synapse development.

# Materials and Methods

### *C. elegans* genetics

Strains were maintained on NGM agar plates supplemented with cholesterol (>99% pure, C8667; Sigma-Aldrich) at 10 µg/ml (normal

cholesterol) or 40 μg/ml (high cholesterol) and seeded with *Escherichia coli* OP50 at 20°C as described previously ([88]). Under 0 μg/ml (low cholesterol) growth conditions, cholesterol was omitted in NGM preparations. The progeny of L4s transferred from normal plates to plates supplemented with low or high cholesterol was considered as F1 generation. Table S1 lists all the transgenes and strains in this study.

We followed standard procedures to generate double/triple mutants and extra-chromosomal arrays ([89]). Expression constructs listed in Table S2 were usually injected at 20 ng/μl with a co-injection marker *Pttx-3::RFP* at 50 ng/μl and pBlueScript at 30 ng/μl. Two to three independent transgenic lines were analyzed for each construct as listed in Table S1.

## Molecular biology and plasmid construction

We cloned the *ncr-1* gene from its genomic locus including 116-bp upstream sequences before the start codon and 847-bp downstream sequences after the stop codon using primer pair 5′-GTTTCATTGAACCGTGTGCT-3′ and 5′-AACTCCGAATAACTTTGGTGCT-3′. We cloned the *ncr-2* gene from its genomic locus including 3-bp upstream sequences before the start codon and 400-bp downstream sequences after the stop codon using primer pair 5′-GGAATGCGTCAAGGAGGAG-3′ and 5′-TGAGCGGGCTAAGACTATTCTG-3′. These two genes were initially subcloned into pCR8 using kit (pCR8/GW/TOPO TA Cloning Kit, #K2500-20; Invitrogen). Expression vectors of *ncr-1* and *ncr-2* driven by *F25B3.3*, *col-10*, *col-19*, or *dpy-7* promoters were generated using Gateway cloning with appropriate destination vectors (LR Clonase II Enzyme Mix, #11791-020; Invitrogen). Two or multiple fragments were assembled into other expression vectors of *ncr-1* and *ncr-2* using Gibson Assembly Master Mix (#E2611S; NEB) ([90]) or NovoRec Plus One-step PCR Cloning Kit (#NR005; Novoprotein). We confirmed all plasmids by sequencing.

Vector *pCR8::ncr-1::gfp* (pFC63) or *pCR8::ncr-2::gfp* (pFC64) was assembled of partial sequences from strain FCZ210 (*ncr-1(cfu39 [ncr-1::GFP])*) or FCZ216 (*ncr-2(cfu45[ncr-2::GFP])*) genome, partial sequences from *pCR8::ncr-1* (pFC43) or *pCR8::ncr-2* (pFC44), and digestion products of Xba I–treated pFC43 or Xho I–treated pFC44 using NovoRec Plus One-step PCR Cloning Kit. *F25B3.3p::ncr-1::gfp* (pFC65), *dpy-7p::ncr-1::gfp* (pFC90), *F25B3.3p::ncr-2::gfp* (pFC66), and *col-10p::ncr-2::gfp* (pFC84) were generated using Gateway cloning. All the expression constructs and related transgenic lines are listed in Table S2.

## Generation of deletion or GFP knock-in alleles by CRISPR/Cas9-mediated genome editing

To make deletion alleles of *sms-1*, *sms-3*, and *sms-5*, we generated P*eft-3-cas9*-NLS-pU6-sgRNA vectors carrying the sgRNAs targeting each gene by site-directed mutagenesis of pDD162 (#47549; Addgene) ([91]). For each gene, 2-4 sgRNAs were designed and mixed with a selection marker pRF4 (*rol-6(su1006)*, with a roller phenotype) at a concentration of 50 ng/μl each for injection. We injected the mixture of plasmids into >20 1-d adults (P0), and singled roller F1 worms on newly seeded OP50 plates 3 d later. The F1

heterozygous deletion alleles were identified using single-worm PCR after enough F2 worms were hatched. Primers used are listed in Table S3.

To make GFP knock-in lines of *ncr-1* and *ncr-2*, we generated P*eft-3-cas9*-NLS-pU6-sgRNA vectors carrying the sgRNAs targeting *ncr-1* and *ncr-2* by site-directed mutagenesis of pDD162. At least two sgRNAs around each targeting site were designed and tested for efficiency. To provide templates for recombination, we cloned 2-kb homology arms on both sides of each target site, which were then assembled with digestion products of *Spe* I (#3133S; NEB)– and *Cla* I–treated (#R0197S; NEB) pDD282 (#66823; Addgene) by Gibson Assembly Master Mix ([90]). Primers used are listed in Table S3. Final homology sequence–containing plasmids have the self-excising drug selection cassette as described previously ([92]).

We co-injected a sgRNA vector (50 ng/μl), a homology sequence–containing vector (35 ng/μl), and marker vectors (10 ng/μl *Prab-3::mcherry*, 5 ng/μl *Pmyo-3::mcherry*, and 2.5 ng/μl *Pmyo-2::mcherry*) into >60 1-d adults (P0). After being maintained at 25°C for 3 d, each 6-cm plate was treated with 500 μl hygromycin (5 mg/ml, 10687010; Invitrogen). Upon starvation, roller worms without any mCherry co-injection markers were singled and screened for 100% roller progeny. To excise the drug selection cassette, we picked 8–10 L1-L2 worms from each potential knock-in plate for heat shock at 34°C for 4 h. Then, non-roller progeny were singled for genotyping the insertion of *gfp* with primers listed in Table S3.

## Fluorescence microscopy

1-d young adults were paralyzed in M9 buffer with 0.5-1% 1-phenoxy-2-propanol (TCI America) before being observed under an Olympus motorized BX53 microscope. The synaptic punctate numbers were scored from the entire dorsal nerve cord of animals with *juIs1[Punc-25-SNB-1::GFP]*or *nuIs152[Punc-129-SNB-1::GFP]*. We then normalized individual number values to the average punctate number of controls in the same experiment. Ectopic axonal branches were scored from 15 to 19 commissures of young adults with *juIs76[Punc-25-GFP]*. Percentages of neurons with ectopic branches per animal were calculated by dividing the commissure number with branches by the total commissure number scored. The percentage of animals with ectopic branches was calculated by dividing the animal number with branches by the total animal number scored.

To assess synaptic size, we acquired images of synapses situated posterior to the vulva region by z-stack imaging with 0.5-μm intervals using a Zeiss LSM 800 confocal microscope equipped with standard Zeiss filters. Maximum projection images from all sections with fluorescence signal were processed in Image-Pro Plus 6.0 software (Bioimager) to calculate the area of each synaptic puncta above the threshold. Typically, we analyzed more than 35 puncta from each animal and recorded the mean value for each animal. We then normalized the mean values of all the worms to the average size of controls in the same experiment. To evaluate the level of co-localization between presynaptic SNB-1::GFP and postsynaptic GABAR::tagRFP, images of synapses were obtained from different genotypes in the *juIs1[Punc-25-SNB-1::GFP]; KrSi2[Punc-49::unc-49B-tagRFP]* background. From the multiple z-stack sections, one section with the most evident expression of SNB-1::GFP and GABAR::

tagRFP was selected for further analysis. The merged image was opened in IPP 6.0, and a line was drawn along the synaptic region. The intensity of the green and red signals was automatically calculated.

To quantify the size of lipid droplets, we obtained images of lipid droplets in gut cells of middle L4-staged animals using z-stack imaging. The projection images were processed in IPP 6.0 to determine the diameter of all the droplets in int2 cells. Typically, we measured more than 100 lipid droplets per animal and recorded the average diameter for each animal.

### Filipin staining

We performed filipin staining following a previously established protocol with slight modifications (93). Synchronized L4 animals were collected, rinsed 3 times with ice-cold ddH2O, and immersed in RFB solution (160 mM KCl, 40 mM NaCl, 20 mM EGTA, 10 mM Spermidine-HCl, 30 mM Na-PIPES, pH 7.4, and 50% methanol) supplemented with 1% PFA (Cat# 15710; 16% solution from Electron Microscopy Sciences). The samples underwent three freeze–thaw cycles in an ethanol-dry ice bath and were subsequently rinsed with PBS to remove PFA. The worms were resuspended in RFB solution containing 625 ng/ml filipin III (Cat# 480-49-9; GLPBIO), along with 0.625% $\beta$-mercaptoethanol and 0.25% phenoxypropanol, flushed with nitrogen, and incubated for 4 h in an ice-water bath shielded from light. After being washed 3 times with PBS, 5 $\mu$l of worm pellet was transferred onto a glass slide, and 5 $\mu$l of antifade mountant (ProLong Diamond Antifade Mountant, P36961) was added. The filipin-stained animals were observed using the Olympus motorized BX53 microscope with the same filter set employed for DAPI staining. The pharynx of each worm was delineated, and the mean intensity was calculated using IPP 6.0.

### Oil Red O staining

We performed Oil Red O staining following the previous protocols (94, 95) with some minor modifications. Middle L4-staged animals were collected and briefly washed with 0.1% Triton X-100/M9 solution. The samples were then washed 3 times with M9 and fixed in 4% PFA for 15 min. Later, the samples were washed again briefly with 0.1% Triton X-100/M9 solution and dehydrated in 60% isopropanol for 2 min after being washed 3 times with M9. The fresh Oil Red O working solution was prepared by mixing 60% volume of stock solution (O1391-250 ml, 0.5% in isopropanol; Sigma-Aldrich) with 40% volume of water and allowing it to equilibrate for 10 min. The mixture was then filtered through a 0.22-$\mu$m syringe filter. The samples were stained with the filtered mixture at room temperature for 60 min with gentle rocking. After staining, the samples were briefly washed twice with 0.1% Triton X-100/M9 solution and thrice with M9. Finally, the stained samples were mounted onto agar-padded slides for imaging using an Olympus MVX10 microscope equipped with a DP23 camera. To determine the Oil Red O signals, the RGB images were first converted to grayscale and then inverted. Fiji software was used to measure the total intensity of the entire worm.

### C. elegans synchronization and lipid profiling

8–16 L4 worms from four strains (juIs1, ncr-1(nr2022); juIs1, ncr-2(nr2023); juIs1, and ncr-1(nr2022); ncr-2(nr2023); juIs1) were transferred to each 9-cm agar plate seeded with OP50. 5 d later, the gravid hermaphrodites were collected in a 50-ml tube and lysed with bleach. Eggs were washed three times with M9, and around 10,000 eggs were transferred to each 9-cm agar plate with OP50. After another 2 d, L4 larvae were harvested in a 15-ml tube at room temperature and washed extensively with M9 to get rid of OP50. We gently discarded the supernatant after worms settled down and counted the worms per microliter. Approximately 10,000 L4 worms were aliquoted, frozen immediately in liquid nitrogen, and kept under −80°C until processed. Lipid extraction and lipidomic analyses were conducted at LipidALL Technologies (www.lipidall.com). Briefly, lipids were extracted using chloroform:methanol and dried in the SpeedVac before being analyzed by ExionLC-AD coupled with Sciex QTRAP 6500 PLUS. The total lipid molecule content in each sample was normalized by dividing it by the respective number of worms. Four biological replicates from each strain were used for analyses. To determine how a lipid or total lipids differed in any two strains, post hoc uncorrected Dunn's test for multiple comparison was conducted after the Kruskal–Wallis test.

Lipids were extracted from ~10,000 of synchronized L4 worms using a modified version of Bligh and Dyer's method as described previously (96). Worms were homogenized in 750 $\mu$l of chloroform: methanol:Milli-Q H2O (3:6:1) (vol/vol/v). The homogenate was then incubated at 25$g$ for 1 h at 4°C. At the end of the incubation, 350 $\mu$l of deionized water and 250 $\mu$l of chloroform were added to induce phase separation. The samples were then centrifuged, and the lower organic phase containing lipids was extracted into a clean tube. Lipid extraction was repeated once by adding 450 $\mu$l of chloroform to the remaining aqueous phase, and the lipid extracts were pooled into a single tube and dried in the SpeedVac under the OH mode. Samples were stored at -80°C until further analysis.

Lipidomic analyses were performed using ExionLC-AD coupled with Sciex QTRAP 6500 PLUS as reported previously (97). Separation of individual lipid classes of polar lipids by normal-phase (NP) HPLC was carried out using a TUP-HB silica column (i.d. 150 × 2.1 mm, 3 $\mu$m) with the following conditions: mobile phase A (chloroform: methanol:ammonium hydroxide, 89.5:10:0.5) and mobile phase B (chloroform:methanol:ammonium hydroxide:water, 55:39:0.5:5.5). MRM transitions were set up for comparative analysis of various polar lipids. Individual lipid species were quantified by referencing spiked internal standards. CL MIX, PA-C 17:0, PC-d31 16:0/18:1, PE-d31 16:0/18:1, d7-PI 15:0/18:1, PS-d31 16:0/18:1, and C12-SM were obtained from Avanti Polar Lipids. Free fatty acids were quantitated using d31-16:0 (Sigma-Aldrich) and d8-20:4 (Cayman Chemicals).

Glycerol lipids including diacylglycerols (DAGs) and triacylglycerols (TAGs) were quantified using a modified version of reverse-phase HPLC/MRM (98). Separation of neutral lipids was achieved on a Phenomenex Kinetex C18 column (i.d. 4.6 × 100 mm, 2.6 $\mu$m) using an isocratic mobile phase containing chloroform: methanol: 0.1 M ammonium acetate (100:100:4) (vol/vol/v) at a flow rate of 300 $\mu$l for 10 min. Levels of short-, medium-, and long-chain TAGs were calculated by referencing spiked internal standards of TAG 14:0 3-d5, TAG 16:0 3-d5, and TAG 18:0 3-d5 obtained from CDN

isotopes, respectively. DAGs were quantified using d5-DAG 17:0/17:0 and d5-DAG 18:1/18:1 as internal standards (Avanti Polar Lipids).

Free cholesterols and cholesteryl esters were analyzed under the atmospheric pressure chemical ionization (APCI) mode on a Jasper HPLC coupled to Sciex 4500 MD as described previously, using d6-cholesterol and d6-C18:0 cholesteryl ester (CE) (CDN isotopes) as internal standards (99).

### 25-NBD cholesterol accumulation assay

The 25-NBD cholesterol feeding medium was prepared according to the previous studies with minor modifications (49). A total volume of 400 $\mu$l of a fresh OP50 culture mixed with either 10 or 50 $\mu$l of 1 mg/ml 25-NBD cholesterol (810250P; Avanti Polar Lipids), dissolved in ethanol, was distributed uniformly onto NGM plates that were devoid of cholesterol. The bacterial lawn was grown overnight at 20°C and used for experiments within a period of 7 d. Any unused plates were stored in the dark at 20°C. Various strains of L4 animals, grown on standard NGM plates, were placed on the lawns, and these animals were designated as the P0 generation. 1-d adult animals of the F3 generation or later, grown in the presence of 25-NBD cholesterol throughout their lifespan, were selected for imaging using the confocal microscope after being washed with M9 for 1 h.

### Brood size assay

10 well-fed L4 hermaphrodites of each genotype were transferred to fresh NGM plates seeded with OP50 and maintained at 20°C. After 2 d, their offspring were killed and counted every day until the parental animals stopped laying eggs. All experiments were repeated at least three times.

### Statistical analyses

Statistical analyses were performed in GraphPad Prism 8.0 (GraphPad Software). The $P$-value between 2 samples was determined using a two-tailed $t$ test if the samples followed the normal distribution or by the Mann–Whitney test if not. Statistical analysis of multiple samples was performed using one-way ANOVA if all samples followed the normal distribution or by the Kruskal–Wallis test if not. To determine how two specific samples from multiple samples differed, post hoc corrected Bonferroni's or Dunn's test for multiple comparison was conducted after one-way ANOVA or Kruskal–Wallis's test.

## Supplementary Information

## Acknowledgements

We thank Drs. Xun Huang, Mei Ding, Mengqiu Dong, Di Chen, Guangshuo Ou, Yingchuan Billy Qi, Pingsheng Liu, Bin Liang, Shaobing Zhang, and Haijun Tu for sharing reagents; and Drs. Zhiping Wang, Dong Yan, Jianke Gong, Dong Lin, Zehua Wang, and Feng Cai for technical suggestions. We thank Drs. Lin Mei, Zhenge Luo, Xun Huang, Mei Ding, Zhiping Wang, Yingchuan Billy Qi, and Yishi Jin for critically reading the article and comments. We acknowledge WormBase for information resource. Some strains were provided by the CGC, which is funded by NIH Office of Research Infrastructure Programs (P40 OD010440). This work was supported by the National Natural Science Foundation of China (Nos. 31970922 and 32160187) to F Chen.

## Author Contributions

A Guo: data curation, formal analysis, validation, investigation, methodology, and writing—original draft.

Q Wu: data curation, formal analysis, validation, investigation, methodology, and writing—original draft.

X Yan: writing—original draft, review, and editing.

K Chen: data curation, formal analysis, investigation, and methodology.

Y Liu: formal analysis and investigation.

D Liang: investigation and methodology.

Y Yang: investigation.

Q Luo: resources.

M Xiong: resources.

Y Yu: resources, data curation, and methodology.

E Fei: resources, data curation, supervision, methodology, and writing—original draft, review, and editing.

F Chen: conceptualization, resources, data curation, formal analysis, supervision, funding acquisition, investigation, visualization, methodology, project administration, and writing—original draft, review, and editing.

## Conflict of Interest Statement

The authors declare that they have no conflict of interest.

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
