## [Reviewer comments · Life Science Alliance]

Life Science Alliance

Differential roles of lysosomal cholesterol transporters in the development of *C. elegans* NMJs

Amin Guo, Qi Wu, Xin Yan, Kanghua Chen, Yuxiang Liu, Dingfa Liang, Yuxiao Yang, Qunfeng Luo, Mingtao Xiong, Yong Yu, Erkang FEI, and Fei Chen

DOI: <https://doi.org/10.26508/lsa.202402584>

Corresponding author(s): Fei Chen, Nanchang University

Review Timeline:

Submission Date:	2024-01-10
Editorial Decision:	2024-02-26
Revision Received:	2024-07-14
Editorial Decision:	2024-07-16
Revision Received:	2024-07-21
Accepted:	2024-07-22

Transaction Report:

February 26, 2024

Re: Life Science Alliance manuscript #LSA-2024-02584-T

Dr. Fei Chen
Nanchang University
China

Dear Dr. Chen,

Thank you for submitting your manuscript entitled "Differential roles of lysosomal cholesterol transporters in synapse development by controlling cholesterol homeostasis" to Life Science Alliance. The manuscript was assessed by expert reviewers, whose comments are appended to this letter. We invite you to submit a revised manuscript addressing the Reviewer comments.

Thank you for this interesting contribution to Life Science Alliance. We are looking forward to receiving your revised manuscript.

Sincerely,

B. MANUSCRIPT ORGANIZATION AND FORMATTING:

Reviewer #1 (Comments to the Authors (Required)):

The authors studied links between cholesterol homeostasis and synapse development using neuromuscular junctions in *Caenorhabditis elegans* as experimental model. Notably, *C. elegans* is sterol auxotroph allowing to modify its sterol content by external supply. As molecular targets, they chose *ncr-1* and *ncr-2*, orthologs of the mammalian Niemann-Pick type C1 protein. This component mediates the exit of cholesterol from the endosomal-lysosomal system in cells and specific variants cause a rare and ultimately fatal lysosomal storage disorder presenting progressive neurovisceral symptoms. The authors report specific changes in total levels of esterified and free cholesterol and sphingomyelin and - albeit small - changes in the number of neuromuscular junctions in animals lacking either or both genes that can be rescued partially with external supply of cholesterol. The authors further see no impact of enzymes mediating synthesis and degradation of sphingomyelin on junction formation. Moreover, they report the expression of the genes in various worm tissues and show that the expression level is influenced by availability of cholesterol. They show evidence for the presence of *ncr-1* and *ncr2* in lysosomes and they present results in supplementary figures on a screen searching for genes mediating an impact of cholesterol homeostasis on synapses. One should note that a few previous studies have addressed the expression pattern of the proteins and the phenotype, notably the Dauer-like larval state in animals lacking both proteins. The new study provides some new insight with respect to the link between cholesterol and the formation of neuromuscular junctions, but interest in the manuscript is diminished by several weaknesses with respect to form, experimental design, and interpretation.

- Pgs. 1-2: The title and abstract should mention the species used in the study.
- Pgs. 2-6. In the Introduction, for example on pg 5 para 1, the authors should cite previous studies examining *ncr1* and *ncr2* in *C. elegans* to indicate that this animal model has previously been used to study the function of the NPC1-like proteins.
- Pg. 5, para 2: The term "less vibrant synapses" is ill-defined and should be replaced.
- Pg. 6, para 2, Fig. 1A; pg. 8, para 2; Fig. 2: The generation of mutant worms is not well explained. Authors state somewhat cryptically "we utilized null allele of *ncr-1*(nr2022), which results in the deletion of multiple exons (Fig. 1A)". They must explain in more detail how these lines were produced. Do they correspond to the GFP-knock-in mentioned on pg. 18 in the Material Methods, or are they different? Is "*ncr-1*(0)" the same as "*ncr-1*(nr2022)"? Similarly, the null mutants *ncr2* should be explained more precisely.
- Results: The authors' bulk analysis of lipids is new and potentially of interest. However, there are several issues that need to be addressed:
 - - Pg. 6, para 2: The term "relatively lower levels of free cholesterol" is a bit unclear and should be replaced.
 - - Pg. 6 and 28, Figs. 1B-D, S2: The abundance of lipid molecules was normalized to "per worm". Is this appropriate assuming that individual worms differ in size and weight? The legend states $n = 4$ biological samples. Does this refer to 4 worms? The authors must clarify this, and provide appropriate normalisation of lipid abundance.
 - - Pg. 6, para 2: The authors' conclusion that "NCR1 promotes the utilization of cholesterol esters and the absorption of cholesterol" is a bit unclear. What is meant by "utilisation" and what is meant by "absorption"? Evidently, esters serve to store and transport cholesterol. How does *C. elegans* acquire cholesterol? By physico-chemical absorption or by receptor-mediated uptake (see also their results with respect to lysosomal location of the *ncr1* and *ncr2*).
 - - The authors' results raise the question in which tissues and cells the changes in cholesterol esters and free cholesterol occur. This should be addressed using histochemical staining of mutant worms with neutral lipid stains (nile red and sudan black) and filipin. The authors should also discuss their findings with respect to the previous work by Wustner et al. (2010 Traffic), who did not detect changes in sterol distribution in *ncr1* mutants.
 - - Figs. 1CD 2CD: the redundant display of free and esterified cholesterol levels in *ncr-1* mutants in panels Fig. 2CD (taken from Fig 1CD) should be avoided. Instead, the authors should include in Fig. 2CD the levels in double mutants.
- Results: Throughout the ms, the authors used different cholesterol concentrations to modify the cholesterol content of the worms and to study the impact on NMJs or on *ncr-1* *ncr-2* levels. However, it is unclear how much of the cholesterol provided externally is internalized. Therefore, control experiments showing the cholesterol content of worms exposed to the different external cholesterol concentrations should be provided to validate that higher cholesterol supply also leads to higher cholesterol levels or fails to do so for example if *ncr-1* is missing.
- Results: The effects on synapses - or rather NMJs - are rather small in the <10% range, therefore, their relevance is unclear. Based on the data shown, one could conclude that NMJ formation is rather independent from external cholesterol or the *ncr-1* or *ncr-2* proteins. It is unclear why the authors state throughout the ms that there are increases or decreases of synapse numbers while the data show really minor changes. This is not supported by the data. In any case, the analyses of synapses show weak points that need to be addressed:
 - - Throughout the ms: the term synapses should be replaced by neuromuscular junctions since the authors studied this special

type of connection.

- - In many plots where bars represent means, authors should include the individual data points showing their distribution. If there are many data points, violin or box plots may be displayed.
- - On pg. 6, para 3, Fig. 1 the authors should properly introduce their analysis of NMJs notably to non-expert readers. What kind of motoneurons are these? The construct used to label GABAergic connections (*unc-25p::SNB-GFP*) should be already mentioned here instead of pg 9 para 2, and a proper reference should be cited. The micrographs shown in Fig. 1E and I are a bit cryptic. Where are these NMJs located in the animal? This should be illustrated properly. How many of these NMJs were analysed per animal?
- - The analyses rely solely on a presynaptic marker to identify NMJs. Therefore, it is not clear whether the puncta they detect are in fact NMJs or simply packages of synaptic vesicles transported along axons of motoneurons. Here, a necessary control experiment is to label in addition a postsynaptic component and to show coincidence of pre- and postsynaptic marker to validate the presence of a NMJ.
- - In Fig. 1, the authors expose three parameters that are in fact highly correlated. A higher intensity of presynaptic marker staining also means more presumable synapses are detected, and their area is bigger. A suggestion here is to skip the intensity parameter (Figs. 1H, 2H) and display only the size. As mentioned, the number of NMJs should be validated by coincidence of pre- and postsynaptic markers.
- Pg. 8, para 1; Fig. 1K: The low number of samples analysed in *Ncr-1* mutant animals fed with low cholesterol raises the question about the viability of these animals. Is the low number of NMJs caused by impaired health? Do these animals have normal life-spans? Reduced viability was shown already by Sym et al. 2000 *Curr Biol*.
- Pgs 11-12, Fig. 3: The data shown in this block seem misplaced here, as the following part (pg. 12, para 2 etc.) is again about synapses. The authors should consider to rearrange the Results. Importantly, the description, analyses and display of data related to *ncr-1* and *ncr-2* expression and the impact of cholesterol are difficult to understand. This should be modified. In Fig. 3F, it is unclear what the puncta mean, and in panels F and G of Fig 3, the cholesterol concentrations used for the different conditions are unclear, instead of "normal" and "low" the concentrations should be indicated? Is there an alternative way to quantify the expression level of *ncr-1* or *ncr-2* using continuous intensities rather than categorical percentages which seems prone to errors?
- Some of the supplementary figures (e.g. Figs. S2, 3, 5) could be shown as main figures.
- Fig. 5D: "Cholesterol Easter" should be corrected.

Reviewer #2 (Comments to the Authors (Required)):

- 1) The author's present a study of the roles of 2 cholesterol transporters of *C. elegans* in synapse formation in this organism, one enhancing this process and the other inhibiting it.
- 2) I believe all the experimental results confirm the conclusions although, not being an expert on this organism, some diminished responsibility for this conclusion should be allowed.
- 3) While experimentally this is an excellent study with sound results, its discussion is misleading, at least to this mammalian geneticist. I believe the statement from the abstract that "NCR-1, a hypothetical lysosomal cholesterol transporter," is quite incorrect. NCR-1 is a homolog of NPC1, a highly studied cholesterol transporter which functions in the same manner even in yeast! In this light, the interesting role of NCR-2 with its opposite effects from NCR-1, and which clearly is not a homolog of NPC-2, but might be a homolog of NPC1L1, a cholesterol transporter involved in cholesterol absorption by enterocytes in the intestine, deserves attention. Does it, too, have a homolog in yeast?
The discussion of NPC1 and its role in mammalian neurons is, in this reviewer's mind somewhat incomplete but not badly so. I believe that NPC-1 is mostly nonautonomous, not sometimes (see Marshall CA, Watkins-Chow DA, Palladino G, Deutsch G, Chandran K, Pavan WJ, Erickson RP (2017) In Niemann-Pick C1 mouse models, glial-only expression of the normal gene extends survival much longer than do changes in genetic background or treatment or treatment with hydroxypropyl-beta-cyclodextrin. *Gene*,643 ;117-123.for instance).

Reviewer #3 (Comments to the Authors (Required)):

In this manuscript Guo and coworkers try to address the role of NPC1 homologs in cholesterol homeostasis and synaptic formation in a *C. elegans* model. Combining lipidomic analysis, with genetic mutant analysis, their results suggest that NCR-1 may act as a lysosomal cholesterol transporter, promoting synaptic development. Loss of *ncr-1* results in smaller synapses and low cholesterol exacerbates the deficits. Unexpectedly, NCR-2, the NCR-1 homolog, seems to increase the utilization of cholesterol and sphingomyelins and impedes synapse formation. NCR-2 deficiency causes an increase in synapses regardless of cholesterol concentration. Inhibiting the degradation or synthesis of sphingomyelins can induce or suppress the synaptic phenotypes in *ncr-2* mutants. Moreover the authors demonstrated that both NCR-1 and NCR-2 are associated with lysosomes and respond to low cholesterol feeding.

These results are interesting but somehow controversial, and further experiments should be performed before these results can be published:

As far as NCR1 is concerned it is unclear why the knock-out of an NPC1 homolog would decrease cholesterol levels and increased cholesterol esterification. Authors should try to understand if the increase in cholesterol ester are due to an increase in ACAT-1 activity and increase in lipid droplets formation, or due to a altered lysosomal function, which could lead to a decrease in lysosomal hydrolase activity. They could label lipid droplets and use a *mboa-1* mutant strain. It would be important to determine if NCR1 knockout leads to any changes in intracellular cholesterol localization, namely lysosomal accumulation. Authors can use either filipin, domain 4 (D4) of Perfringolysin O, etc. Moreover, it is not clear if changes in cholesterol levels are occurring in neurons at the synaptic levels. Authors can isolate synaptosomes, and if this is impossible in *C. elegans*, at least isolate neurons and quantify cholesterol levels. They can alternatively colocalize changes in filipin fluorescence with synaptic markers. Moreover, it would be interesting to determine if NCR1 decreases 7-dehydrocholesterol levels.

As far as NCR2 is concerned, it would also be interesting to determine some of the previous aspects (intracellular cholesterol alterations/ lipid droplet formation) and determine if the inhibition of *daf-36* could have a similar effect, suggesting, that indeed, in NCR2 KO there is a rerouting cholesterol metabolism. Here again quantification of cholesterol in synapses would be very important.

Authors' response:

We extend our gratitude to all the reviewers for their time and effort in evaluating our manuscript. We have thoroughly addressed all the concerns raised by the reviewers in the following sections.

(Reviewer's comments for the Author are shown *italic*, and our response in blue)

Reviewer #1 (Comments to the Authors (required)):

*The authors studied links between cholesterol homeostasis and synapse development using neuromuscular junctions in *Caenorhabditis elegans* as experimental model. Notably, *C. elegans* is sterol auxotroph allowing to modify its sterol content by external supply. As molecular targets, they chose *ncr-1* and *ncr-2*, orthologs of the mammalian Niemann-Pick type C1 protein. This component mediates the exit of cholesterol from the endosomal-lysosomal system in cells and specific variants cause a rare and ultimately fatal lysosomal storage disorder presenting progressive neurovisceral symptoms. The authors report specific changes in total levels of esterified and free cholesterol and sphingomyelin and - albeit small - changes in the number of neuromuscular junctions in animals lacking either or both genes that can be rescued partially with external supply of cholesterol. The authors further see no impact of enzymes mediating synthesis and degradation of sphingomyelin on junction formation. Moreover, they report the expression of the genes in various worm tissues and show that the expression level is influenced by availability of cholesterol. They show evidence for the presence of *ncr-1* and *ncr2* in lysosomes and they present results in supplementary figures on a screen searching for genes mediating an impact of cholesterol homeostasis on synapses. One should note that a few previous studies have addressed the expression pattern of the proteins and the phenotype, notably the Dauer-like larval state in animals lacking both proteins. The new study provides some new insight with respect to the link between cholesterol and the formation of neuromuscular junctions, but interest in the manuscript is diminished by several weaknesses with respect to form, experimental design, and interpretation.*

- Pgs. 1-2: *The title and abstract should mention the species used in the study.*

Ans: We apologize for the confusion and have made the following revisions (marked up in the manuscript).

On Pg. 1 Title: "Differential roles of lysosomal cholesterol transporters in the development of *C. elegans* neuromuscular junctions by controlling cholesterol homeostasis"

On Pg.2 Abstract: "Cholesterol homeostasis in neurons is critical for synapse formation and maintenance. Neurons with impaired cholesterol uptake undergo progressive synapse loss and eventual degeneration. To investigate the molecular mechanisms of neuronal cholesterol homeostasis and its role during synapse development, we studied motor neurons of *C. elegans* because these neurons rely on dietary cholesterol."

- Pgs. 2-6. *In the Introduction, for example on pg 5 para 1, the authors should cite previous studies*

examining *ncr1* and *ncr2* in *C. elegans* to indicate that this animal model has previously been used to study the function of the NPC1-like proteins.

Ans: We have included a paragraph on Pg. 5 to introduce *ncr-1* and *ncr-2* in *C. elegans*.

"Following endocytosis, cholesteryl esters undergo hydrolysis in lysosomes, yielding unesterified cholesterol that is subsequently exported to various cellular compartments (Meng et al., 2020). The NPC1 protein, a multiple transmembrane protein located on LE/Ly, is primarily responsible for the egress of cholesterol from lysosomes to other cellular compartments in mammals (Carstea et al., 1997; Kwon et al., 2009; Loftus et al., 1997). The *C. elegans* genome encodes two NPC1 homologs, *ncr-1* and *ncr-2*, known to participate in intracellular cholesterol processing, hormone production and reproductive growth (Li et al., 2004; Motola et al., 2006; Sym et al., 2000; Yochem et al., 1999). In this study, we first examined cholesterol levels and neuromuscular junctions (NMJs) in *ncr-1* and *ncr-2* null mutants."

- Pg. 5, para 2: The term "less vibrant synapses" is ill-defined and should be replaced.

Ans: We have deleted "and less vibrant".

- Pg. 6, para 2, Fig. 1A; pg. 8, para 2; Fig. 2: The generation of mutant worms is not well explained. Authors state somewhat cryptically "we utilized null allele of *ncr-1(nr2022)*, which results in the deletion of multiple exons (Fig. 1A)". They must explain in more detail how these lines were produced. Do they correspond to the GFP-knock-in mentioned on pg. 18 in the Material Methods, or are they different? Is "*ncr-1(0)*" the same as "*ncr-1(nr2022)*"? Similarly, the null mutants *ncr2* should be explained more precisely.

Ans: We apologize for the confusion. Null alleles of *ncr-1(nr2022)* and *ncr-1(nr2023)* were previously generated in studies conducted by Liu et al. (1999), and we acquired these two alleles from CGC. These two alleles harbor large deletions rendering them nonfunctional, thus classified as null alleles. We denoted *ncr-1(nr2022)* and *ncr-2(nr2023)* as *ncr-1(0)* and *ncr-2(0)*. We have also made the following revisions to the manuscript.

On Pg. 6, para 2:

"To investigate the potential role of NCR-1 in maintaining cholesterol homeostasis and synapse development, we utilized *ncr-1(nr2022)*, a mutant obtained via an EMS screen (Liu et al., 1999). The *ncr-1(nr2022)* allele results in the deletion of multiple exons, rendering it null, and is therefore denoted as *ncr-1(0)* (Fig. 1A)."

On Pg. 8, para 3:

"Null mutants of *ncr-2*, resulting from the *nr2023* deletion allele (Liu et al., 1999) (Fig. 2A), exhibited no significant alterations in the overall lipid levels in the 10 subcategories (TAGs, DAGs, FFAs, PCs, PAs, PSs, Pls, PEs, CLs and PGs) (Supplemental Fig. S3C-S3L)."

- Results: The authors' bulk analysis of lipids is new and potentially of interest. However, there are several issues that need to be addressed:

- - Pg. 6, para 2: The term "relatively lower levels of free cholesterol" is a bit unclear and should be replaced.

Ans: We apologize for the confusion. We have deleted "relatively" in the manuscript.

- - Pg. 6 and 28, Figs. 1B-D, S2: The abundance of lipid molecules was normalized to "per worm". Is this appropriate assuming that individual worms differ in size and weight? The legend states $n = 4$ biological samples. Does this refer to 4 worms? The authors must clarify this, and provide appropriate normalisation of lipid abundance.

Ans: We apologize for the confusion. Detailed information on lipid extraction and lipidomic analyses is provided in the supplemental methods section. We synchronized the worm cultures and collected 10000 L4 worms from each strain for each batch of samples. In total, 4 batches of samples were prepared for lipidomic analyses. Synchronized L4 worms showed reduced variation in size and weight compared to mixed-staged worms. To normalize, the total lipid molecule content in each sample was divided by the corresponding number of worms.

We have provided information on normalization procedures in the Materials and Methods section.

On Pg. 21 in Materials and Methods: "Briefly, lipids were extracted using chloroform:methanol and dried in the SpeedVac before analyzed by a ExionLC-AD coupled with Sciex QTRAP 6500 PLUS. The total lipid molecule content in each sample was normalized by dividing it by the respective number of worms. Four biological replicates from each strain were used for analyses."

- - Pg. 6, para 2: The authors' conclusion that "NCR1 promotes the utilization of cholesterol esters and the absorption of cholesterol" is a bit unclear. What is meant by "utilisation" and what is meant by "absorption"? Evidently, esters serve to store and transport cholesterol. How does *C. elegans* acquire cholesterol? By physico-chemical absorption or by receptor-mediated uptake (see also their results with respect to lysosomal location of the *ncr1* and *ncr2*).

Ans: We apologize for the confusion. We have revised this sentence.

On Pg. 7, para 1:

"the *ncr-1(0)* mutants exhibit higher levels of cholesteryl esters and lower levels of free cholesterol compared to the wild type (Fig. 1B and 1C). These results suggest that NCR-1 promotes cholesterol mobilization."

Despite the limited understanding of the molecular mechanisms involved in cholesterol uptake and distribution, research indicates that *C. elegans* acquire cholesterol through receptor-mediated uptake. The expression of the membrane-associated protein CUP-1 in the digestive tract cells is essential for cholesterol absorption in the gut (Valdes et al., 2012). Cholesterol from intestinal cells is co-released with yolk proteins and transported into the oocytes via receptor-mediated endocytosis (Matyash et al., 2001). A gp330/megalyn-related protein, worm LRP-1, is proposed to mediate cholesterol endocytosis in the epidermis (Yochem et al., 1999). Consistent with previous studies (Merris et al., 2003), our filipin staining shows that the fluorescence in the pharynx exhibits

a proximal-to-distal gradient (Fig 4F), supporting that cholesterol is absorbed from the digestive tract. Our results show that the increase of ORO signals and droplet size in the intestine of *ncr-1(0); ncr-2(0)* double mutants is not mediated by *mboa-1* (Fig 4A-4E), the membrane bound O-acyltransferase in *C. elegans*, implying that the increased cholesteryl esters result from compromised hydrolysis of cholesteryl esters. Therefore, both cholesterol and cholesteryl esters may be endocytosed from the digestive lumen. After aiding in hydrolyzing cholesteryl esters in the lysosomes, NCR-1 and NCR-2 can mediate differential export of free cholesterol.

- - *The authors' results raise the question in which tissues and cells the changes in cholesterol esters and free cholesterol occur. This should be addressed using histochemical staining of mutant worms with neutral lipid stains (nile red and sudan black) and filipin. The authors should also discuss their findings with respect to the previous work by Wustner et al. (2010 Traffic), who did not detect changes in sterol distribution in ncr1 mutants.*

Ans: We are grateful to the reviewer for the insightful suggestions. To visualize neutral lipids and free cholesterol, we have performed Oil red O staining and filipin staining in N2, *ncr-1(0)*, *ncr-2(0)* and *ncr-1(0); ncr-2(0)* mutant animals. We have provided the data in the revised Fig 4.

Oil red O primarily stained the intestines of N2 animals, the main site of fat storage in *C. elegans* (Fig 4C). In *ncr-1(0)* and *ncr-2(0)* mutants, while cholesteryl esters were elevated compared to N2 controls, no changes in ORO signals were observed across the body (Fig 4C and 4D). However, in *ncr-1(0); ncr-2(0)* double mutants, which displayed a substantial increase in cholesteryl esters, ORO signals significantly increased (Fig 4C and 4D).

Due to interference from autofluorescence in the intestine, we focused on the pharynx of filipin-stained animals (Fig 4F). Consistent with mass spectrometry data, *ncr-2(0)* mutants exhibited higher filipin fluorescence, while both *ncr-1(0)* and *ncr-1(0); ncr-2(0)* mutants showed lower filipin fluorescence compared to N2 animals (Fig 4F and 4G).

Interestingly, a prior study by Wustner (2010) using DHE as the sterol source, reports no alterations in sterol distribution in worms that were fed with bacteria containing dsRNA against the *ncr-1* genes. We have discussed our findings in relation to prior research (Wustner et al., 2010) within the manuscript.

On Pg. 17 para 3:

".....Both *ncr-1* and *ncr-2* have been implicated in intracellular cholesterol processing, as single or double mutants are unable to grow without cholesterol and their growth defects can be rescued by daifachronic acids derived from cholesterol (Li et al., 2004; Motola et al., 2006; Sym et al., 2000). Our research demonstrates that NCR-1 and NCR-2 exert distinct effects on cholesterol levels. A prior study utilizing dehydroergosterol (DHE), a natural fluorescent analog of cholesterol, did not reveal any changes in sterol distribution in worms fed bacteria containing dsRNA targeting the *ncr-1* genes (Wustner et al., 2010). It is important to note that the outcomes of RNA interference and null mutants may differ. Furthermore, the esterified form of DHE exhibits fluorescence similar to non-esterified DHE, making them indistinguishable under UV-sensitive wide-field microscope. Therefore, the accumulation of cholesteryl esters in worms treated with *ncr-1* dsRNA or lacking *ncr-1* might mask any reduction in cholesterol levels when employing the DHE method....."

- - *Figs. 1CD 2CD: the redundant display of free and esterified cholesterol levels in ncr-1 mutants in panels Fig. 2CD (taken from Fig 1CD) should be avoided. Instead, the authors should include in Fig. 2CD the levels in double mutants.*

Ans: We have revised Fig 2 to eliminate the redundant representation of free and esterified cholesterol levels in ncr-1 mutants. The updated Fig. 2BCD now includes the levels of free and esterified cholesterol, as well as sphingomyelins, in double mutants.

- *Results: Throughout the ms, the authors used different cholesterol concentrations to modify the cholesterol content of the worms and to study the impact on NMJs or on ncr-1 ncr-2 levels. However, it is unclear how much of the cholesterol provided externally is internalized. Therefore, control experiments showing the cholesterol content of worms exposed to the different external cholesterol concentrations should be provided to validate that higher cholesterol supply also leads to higher cholesterol levels or fails to do so for example if ncr-1 is missing.*

Ans: We thank the reviewer for pointing this out. We performed filipin staining using N2 and mutant animals that were maintained under low (0 ng/μl) and normal (10 ng/μl) cholesterol conditions. We have provided the data in the revised Fig 4.

As shown in Fig 3G, N2 animals grown in normal cholesterol conditions exhibited higher filipin fluorescence in the pharynx than those grown in low cholesterol conditions. The ncr-1(0) and ncr-1(0); ncr-2(0) mutants rarely showed significant filipin fluorescence in the pharynx of normally fed animals. Notably, ncr-2(0) mutants, which displayed pronounced filipin fluorescence under normal cholesterol conditions, exhibited markedly reduced fluorescence under low cholesterol conditions. These results strongly suggest that dietary cholesterol affects the cholesterol content in worms.

- *Results: The effects on synapses - or rather NMJs - are rather small in the <10% range, therefore, their relevance is unclear. Based on the data shown, one could conclude that NMJ formation is rather independent from external cholesterol or the ncr-1 or ncr-2 proteins. It is unclear why the authors state throughout the ms that there are increases or decreases of synapse numbers while the data show really minor changes. This is not supported by the data. In any case, the analyses of synapses show weak points that need to be addressed:*

Ans: We appreciate the critical comments from the reviewer. We acknowledge that the synaptic phenotypes induced by ncr-1(0) or ncr-2(0) are subtle. Specifically, in ncr-1(0) mutants, synapse size decreased by 25% compared to control animals, while synapse number increased by less than 10% in ncr-2(0) mutants. Nevertheless, these synaptic phenotypes are consistently reproducible. By placing control and mutant animals on the same slide, we consistently identify ncr-1(0) mutants. We are also intrigued by the contrasting regulatory effects of ncr-1(0) and ncr-2(0) on synapses. Additionally, under both low and high cholesterol conditions, ncr-1(0) and ncr-2(0) mutants exhibit distinct alterations in synapses.

- - *Throughout the ms: the term synapses should be replaced by neuromuscular junctions since the authors studied this special type of connection.*

Ans: We have substituted the term "synapses" with "neuromuscular junctions" or "NMJs" in the manuscript as needed.

- - *In many plots where bars represent means, authors should include the individual data points showing their distribution. If there are many data points, violin or box plots may be displayed.*

Ans: We have incorporated individual data points in each of the bar panels.

- - *On pg. 6, para 3, Fig. 1 the authors should properly introduce their analysis of NMJs notably to non-expert readers. What kind of motoneurons are these? The construct used to label GABAergic connections (*unc-25p::SNB-GFP*) should be already mentioned here instead of pg 9 para 2, and a proper reference should be cited. The micrographs shown in Fig. 1E and I are a bit cryptic. Where are these NMJs located in the animal? This should be illustrated properly. How many of these NMJs were analysed per animal?*

Ans: We apologize for any confusion. The section introducing the transgenic line labeling GABAergic connections has been relocated. Additionally, an illustration (Fig 1E) has been added to depict the NMJs analyzed in our study. In our analysis of synapse number, we typically counted approximately 180 puncta for presynaptic boutons in the dorsal processes of DD type-GABAergic motor neurons, and nearly 280 puncta for the presynaptic boutons in the dorsal processes of DA/DB/AS type-cholinergic motor neurons. For synapse size analysis, we imaged presynaptic boutons posterior to the vulva region, with an average of 22 puncta measured per worm. To minimize experimental variability, normalization procedures were performed as outlined in the Materials and Methods section (Pg. 21).

On Pg. 7, para 2:

"We then examined the synapses between GABAergic motor neurons and muscle cells. Vesicle-associated membrane protein SNB-1-GFP/VAMP, driven by *unc-25* promoter, labels the presynaptic bouton in GABAergic motor neurons (Jin et al., 1999; Nonet, 1999). Our analysis was focused on the presynaptic boutons present in the dorsal processes of DD motor neurons (Fig. 1E)."

- - *The analyses rely solely on a presynaptic marker to identify NMJs. Therefore, it is not clear whether the puncta they detect are in fact NMJs or simply packages of synaptic vesicles transported along axons of motoneurons. Here, a necessary control experiment is to label in addition a postsynaptic component and to show coincidence of pre- and postsynaptic marker to validate the presence of a NMJ.*

Ans: We crossed the transgenic line expressing *unc-25p::SNB-1::GFP* with MosSCI line *KrSi2[unc-49p::unc-49B-tagRFP]*, which labels a component of GABA ionotropic receptors. We performed confocal imaging and observed the exact convergence of *SNB-1::GFP* puncta and *unc-49B::tagRFP* puncta, thereby confirming that synapses are present in regions where *SNB-1::GFP* puncta are located. These results are illustrated in Fig 1F and Fig 2E.

- - In Fig. 1, the authors expose three parameters that are in fact highly correlated. A higher intensity of presynaptic marker staining also means more presumable synapses are detected, and their area is bigger. A suggestion here is to skip the intensity parameter (Figs. 1H, 2H) and display only the size. As mentioned, the number of NMJs should be validated by coincidence of pre- and postsynaptic markers.

Ans: We have deleted the intensity parameter (Figs. 1H, 2H) of synapses in *ncr-1(0)* and *ncr-2(0)* mutants.

- Pg. 8, para 1; Fig. 1K: The low number of samples analysed in *Ncr-1* mutant animals fed with low cholesterol raises the question about the viability of these animals. Is the low number of NMJs caused by impaired health? Do these animals have normal life-spans? Reduced viability was shown already by Sym et al. 2000 *Curr Biol*.

Ans: We agree with you that *ncr-1* mutant animals grown in low cholesterol exhibit reduced viability, as demonstrated by Sym et al. (2000) in *Curr Biol*. In the case of *ncr-1(0)* mutants with NCR-1 overexpression in neurons, although animal viability remains low under low cholesterol conditions, we observed a complete rescue of the synaptic phenotype. This indicates that the severe synaptic defects observed in *ncr-1(0)* mutants under low cholesterol (0 $\mu\text{g/ml}$) are not attributable to impaired health.

- Pgs 11-12, Fig. 3: The data shown in this block seem misplaced here, as the following part (pg. 12, para 2 etc.) is again about synapses. The authors should consider to rearrange the Results. Importantly, the description, analyses and display of data related to *ncr-1* and *ncr-2* expression and the impact of cholesterol are difficult to understand. This should be modified. In Fig. 3F, it is unclear what the puncta mean, and in panels F and G of Fig 3, the cholesterol concentrations used for the different conditions are unclear, instead of "normal" and "low" the concentrations should be indicated? Is there an alternative way to quantify the expression level of *ncr-1* or *ncr-2* using continuous intensities rather than categorical percentages which seems prone to errors?

Ans: We apologize for the confusion. We have rearranged the Results section.

The data illustrating *ncr-1* and *ncr-2* expression and the impact of cholesterol are now presented in Fig S6, supplementing Fig 6. We have replaced the terms "normal" and "low" with the actual cholesterol concentrations used in the NGM plates. Under normal cholesterol concentration (10 $\mu\text{g/ml}$), no NCR-1::GFP expression was observed in the head of any worms under the compound scope. In contrast, under low cholesterol concentration (0 $\mu\text{g/ml}$), NCR-1::GFP expression appeared in the head of over 60% animals (shown in Fig S5D). To minimize variability, each strain was counted at least twice, with a total of more than 175 worms.

- Some of the supplementary figures (e.g. Figs. S2, 3, 5) could be shown as main figures.

Ans: We have shown Figs. S3 and S5 as part of main Figures (Figs. 1I, 1J and 5).

- Fig. 5D: "Cholesterol Easter" should be corrected.

Ans: Corrected.

Reviewer #2 (Comments to the Authors (Required))

1) The author's present a study of the roles of 2 cholesterol transporters of *C. elegans* in synapse formation in this organism, one enhancing this process and the other inhibiting it.
2) I believe all the experimental results confirm the conclusions although, not being an expert on this organism, some diminished responsibility for this conclusion should be allowed.
3) While experimentally this is an excellent study with sound results, its discussion is misleading, at least to this mammalian geneticist. I believe the statement from the abstract that "NCR-1, a hypothetical lysosomal cholesterol transporter," is quite incorrect. NCR-1 is a homolog of NPC1, a highly studied cholesterol transporter which functions in the same manner even in yeast! In this light, the interesting role of NCR-2 with its opposite effects from NCR-1, and which clearly is not a homolog of NPC-2, but might be a homolog of NPC1L1, a cholesterol transporter involved in cholesterol absorption by enterocytes in the intestine, deserves attention. Does it, too, have a homolog in yeast?

Ans: We are grateful for your encouragement. NPC1L1, a homolog of NPC1, is enriched in the brush border membrane of intestinal enterocytes and is critical for the uptake of dietary cholesterol across the plasma membrane (Davies et al., 2000; Altmann et al., 2004; Sane et al., 2006). Although human NPC1L1 has been observed in association with lysosomes (Sane et al., 2006), it lacks distinct tyrosine- or dileucine-based lysosome sorting motif. Comparison of protein sequences reveals that both *C. elegans* NCR-1 and NCR-2 possess lysosome-targeting signal in their C-terminal region (Fig S1 and S2). Our data further confirm the localization of both *C. elegans* NCR-1 and NCR-2 in lysosomes (Fig 6B and 6C). Thus, we propose that both NCR-1 and NCR-2 are homologs of NPC1 and function in the lysosomes. In yeast, the sole NPC1 orthologue reported is the NCR-1, which localizes to the vacuole, analogous to the mammalian lysosome.

We agree to your opinion that NPC1 is a well-studied lysosomal cholesterol transporter. Previous studies have uncovered the roles of *C. elegans* NCR-1 and NCR-2 in reproduction and cholesterol processing. We have omitted the term "hypothetical" from both the abstract and main text.

The discussion of NPC1 and its role in mammalian neurons is, in this reviewer's mind somewhat incomplete but not badly so. I believe that NPC-1 is mostly nonautonomous, not sometimes (see Marshall CA, Watkins-Chow DA, Palladino G, Deutsch G, Chandran K, Pavan WJ, Erickson RP (2017) In Niemann-Pick C1 mouse models, glial-only expression of the normal gene extends survival much longer than do changes in genetic background or treatment or treatment with hydroxypropyl-beta-cyclodextrin. Gene,643 ;117-123.for instance).

Ans: Thank you for your insightful comments. The discussion paragraph has been revised accordingly.

On Pg. 18, para 1,

"NPC1 is expressed in both glia and neurons in the rodent brain (Falk et al., 1999; Prasad et al., 2000). The effects of NPC1 on the survival of Purkinje neurons (PNs) have been considered cell-autonomous. Deletion of *npc1* specifically in PNs causes PN loss in wild type mice (Elrick et al., 2010), while expression of *npc1* specifically in PNs corrects cholesterol accumulation and prevents PN loss in *npc1*^{-/-} mice (Lopez et al., 2011). Interestingly, NPC1 also functions in a non-cell-autonomous manner. For example, astrocyte-specific expression of NPC1 reduces neuronal cholesterol, delays neurodegeneration and extends survival (Zhang et al., 2008; Borbon et al., 2012; Marshall et al., 2017). Moreover, neuron-specific deletion of *npc1* blocks oligodendrocyte maturation and subsequent myelination (Yu and Lieberman, 2013), and cortical neurons co-cultured with NPC1-deficient astrocytes exhibit impaired neurite outgrowth (Chen et al., 2007)."

Reviewer #3 (Comments to the Authors (Required)):

In this manuscript Guo and coworkers try to address the role of NPC1 homologs in cholesterol homeostasis and synaptic formation in a C. elegans model. Combining lipidomic analysis, with genetic mutant analysis, their results suggest that NCR-1 may act as a lysosomal cholesterol transporter, promoting synaptic development. Loss of ncr-1 results in smaller synapses and low cholesterol exacerbates the deficits. Unexpectedly, NCR-2, the NCR-1 homolog, seems to increase the utilization of cholesterol and sphingomyelins and impedes synapse formation. NCR-2 deficiency causes an increase in synapses regardless of cholesterol concentration. Inhibiting the degradation or synthesis of sphingomyelins can induce or suppress the synaptic phenotypes in ncr-2 mutants. Moreover the authors demonstrated that both NCR-1 and NCR-2 are associated with lysosomes and respond to low cholesterol feeding.

These results are interesting but somehow controversial, and further experiments should be performed before these results can be published:

As far as NCR1 is concerned it is unclear why the knock-out of an NPC1 homolog would decrease cholesterol levels and increased cholesterol esterification. Authors should try to understand if the increase in cholesterol ester are due to an increase in ACAT-1 activity and increase in lipid droplets formation, or due to an altered lysosomal function, which could lead to a decrease in lysosomal hydrolase activity. They could label lipid droplets and use a mboa-1 mutant strain.

Ans: We thank the reviewer for pointing this out. We have performed the requested experiments and provided the data in the revised Fig 4.

In *ncr-1(0)* and *ncr-2(0)* mutants, although cholesteryl esters were elevated compared to N2 controls, no changes in ORO signals or the size of lipid droplets were observed (Fig 4A-4D). However, in *ncr-1(0); ncr-2(0)* double mutants, which displayed a substantial increase in cholesteryl esters, both ORO signals and the droplet size significantly increased (Fig 4A-4D). While *mboa-1(tm2464)* mutation reduced ORO signals and droplet size, it did not prevent the increase in these parameters in *ncr-1(0); ncr-2(0)* mutants (Fig 4A, 4B and 4E). These findings suggest that the increase in cholesteryl esters in *ncr-1(0); ncr-2(0)* mutants results from disrupted lysosomal function rather than elevated ACAT-1 activity.

It would be important to determine if NCR1 knockout leads to any changes in intracellular cholesterol localization, namely lysosomal accumulation. Authors can use either filipin, domain 4 (D4) of Perfringolysin O, etc. Moreover, it is not clear if changes in cholesterol levels are occurring in neurons at the synaptic levels. Authors can isolate synaptosomes, and if this is impossible in C. elegans, at least isolate neurons and quantify cholesterol levels. They can alternatively colocalize changes in filipin fluorescence with synaptic markers.

Ans: We appreciate the reviewer's insightful suggestion and agree on the importance of demonstrating changes in subcellular cholesterol distribution in mutant animals.

To visualize the subcellular distribution of free cholesterol, we performed filipin staining on both N2 and mutant animals, as shown in the revised Fig 4. We consistently observed alterations in cholesterol levels in the pharynx of filipin-stained animals, corroborating our mass spectrometry data (Fig 4F). However, imaging of motor neurons and NMJs located in the ventral and dorsal cords was not feasible due to autofluorescence interference from the intestine. Our attempts to analyze the filipin signal in the nerve ring, a structure rich in synapses near the pharynx, were unsuccessful, because the filipin fluorescence in the nerve ring was comparable to the background signal. It is likely that the amount of cholesterol required for neuronal development is low, which makes it difficult to accurately observe free cholesterol using filipin or overexpression of D4 of Perfringolysin O.

Additionally, we attempted to observe the distribution of 25-NBD, a fluorescent analog of cholesterol, by feeding it to worms. Unfortunately, we only observed significant signals in the embryos of adult animals and not in somatic cells (Supplemental Fig. S4).

We also conducted a thorough review of the existing literature but did not find any established protocols for isolating synaptosomes or neurons in *C. elegans* for lipid analysis. This gap in the literature is likely due to the challenges posed by the organism's tough cuticle and the small size of its neurons.

Moreover, it would be interesting to determine if NCR1 decreases 7-dehydrocholesterol levels.

Ans: We agree with the reviewer's opinion that investigating alterations in cholesterol-derived metabolites in mutant animals would be a fascinating area of research. We have consulted our collaborator who pioneered lipidomic analyses of *C. elegans* in China. They told us that they have not yet developed reliable methods to detect the cholesterol-derived metabolites in *C. elegans*. As a result, we are currently unable to examine 7-dehydrocholesterol levels.

As far as NCR2 is concerned, it would also be interesting to determine some of the previous aspects (intracellular cholesterol alterations/ lipid droplet formation) and determine if the inhibition of daf-36 could have a similar effect, suggesting, that indeed, in NCR2 KO there is a rerouting cholesterol metabolism. Here again quantification of cholesterol in synapses would be very important.

Ans: We appreciate your valuable suggestions. We conducted the necessary experiments using *ncr-2(0)* mutants, and the results are provided in the revised Fig 4.

Consistent with lipidomic data, *ncr-2(0)* mutants exhibited the most pronounced filipin fluorescence in the pharynx when compared to N2 and other mutant animals (Fig 4F and 4G). However, we did not observe clear filipin signals from neurons and synapses in *ncr-2(0)* mutants. Despite the lipidomic analysis indicating elevated levels of cholesteryl esters in *ncr-2(0)* mutants, we did not detect any changes in droplet size (Fig 4A). On the other hand, the *daf-36(k114)* mutation alone resulted in a significant increase in the size of lipid droplets and ORO signals (Fig 4A-4D). This discrepancy may be attributed to the smaller increase in cholesterol and cholesteryl esters in *ncr-2(0)* mutants.

July 16, 2024

RE: Life Science Alliance Manuscript #LSA-2024-02584-TR

Dr. Fei Chen
Nanchang University
Qianhu Campus, 999 XueFu Avenue
Nanchang, Jiangxi 330031
China

Dear Dr. Chen,

Thank you for submitting your revised manuscript entitled "Differential roles of lysosomal cholesterol transporters in the development of *C. elegans* NMJs". We would be happy to publish your paper in Life Science Alliance pending final revisions necessary to meet our formatting guidelines.

- please be sure that the authorship listing and order is correct
- please upload all figure files as individual ones, including the supplementary figure files; all figure legends should only appear in the main manuscript files
- please upload your main manuscript text as an editable doc file
- please upload your Tables in editable .doc or Excel format
- please add your main, supplementary figure, and table legends to the main manuscript text after the references section
- please add a Summary Blurb/Alternate Abstract and a Running Title to our system
- please add the Twitter handle of your host institute/organization as well as your own or/and one of the authors in our system
- titles in the system and manuscript file must match
- we encourage you to revise the figure legend for Figure 1 such that the figure panels are introduced in alphabetical order
- Figures S1 and S2 have only one panel, so there is no need to label them as A. Please remove A from the legends
- please add a call-out for Figure S6C to your main manuscript text
- please incorporate the supplemental methods and references into the main Materials & Methods and Reference lists. There is no word limit for these sections.

A. FINAL FILES:

B. MANUSCRIPT ORGANIZATION AND FORMATTING:

Sincerely,

July 22, 2024

RE: Life Science Alliance Manuscript #LSA-2024-02584-TRR

Dr. Fei Chen
Nanchang University
Qianhu Campus, 999 XueFu Avenue
Nanchang, Jiangxi 330031
China

Dear Dr. Chen,

Thank you for submitting your Research Article entitled "Differential roles of lysosomal cholesterol transporters in the development of *C. elegans* NMJs". It is a pleasure to let you know that your manuscript is now accepted for publication in Life Science Alliance. Congratulations on this interesting work.

DISTRIBUTION OF MATERIALS:

Again, congratulations on a very nice paper. I hope you found the review process to be constructive and are pleased with how the manuscript was handled editorially. We look forward to future exciting submissions from your lab.

Sincerely,
